# High COVID-19 transmission potential associated with re-opening universities can be mitigated with layered interventions

Ellen Brooks-Pollock [1,2✉], Hannah Christensen[2], Adam Trickey[2], Gibran Hemani [2], Emily Nixon [3], Amy C. Thomas[1], Katy Turner [1,2], Adam Finn[4], Matt Hickman[2], Caroline Relton[2] & Leon Danon[5]

Controlling COVID-19 transmission in universities poses challenges due to the complex social networks and potential for asymptomatic spread. We developed a stochastic transmission model based on realistic mixing patterns and evaluated alternative mitigation strategies. We predict, for plausible model parameters, that if asymptomatic cases are half as infectious as symptomatic cases, then 15% (98% Prediction Interval: 6–35%) of students could be infected during the first term without additional control measures. First year students are the main drivers of transmission with the highest infection rates, largely due to communal residences. In isolation, reducing face-to-face teaching is the most effective intervention considered, however layering multiple interventions could reduce infection rates by 75%. Fortnightly or more frequent mass testing is required to impact transmission and was not the most effective option considered. Our findings suggest that additional outbreak control measures should be considered for university settings.

[1] Bristol Veterinary School, University of Bristol, Langford, Bristol, UK. [2] Population Health Sciences, Bristol Medical School, University of Bristol, Bristol, UK. [3] School of Biological Sciences, University of Bristol, Bristol, Bristol, UK. [4] Bristol Children's Vaccine Centre, University of Bristol, Bristol, Bristol, UK. [5] Department of Engineering Mathematics, University of Bristol, Bristol, Bristol, UK. ✉email: Ellen.Brooks-Pollock@bristol.ac.uk

Despite the on-going COVID-19 epidemic, social distancing measures in many countries were beginning to be relaxed and universities across the world re-opened for the new academic term in September 2020. In the UK, there are 2.3 million students, with up to 40,000 undergraduates at a single institution[1]. Universities are integral to many towns and cities in the UK: for example, in the 2011 census, a quarter of Oxford's adult population was registered as a full-time student. Managing universities is a complex operation, and in the context of the COVID-19 epidemic, re-opening universities poses particular challenges for containing disease transmission.

In June 2020, the UK moved from a national containment strategy to localised containment of outbreaks, with the number of cases highly variable across the country. The imposition of lockdown in March 2020 led to a substantial reduction in travel and mobility, and local lockdowns have led to further reduced movement in some parts of the country. In the UK, re-opening universities is associated with a mass travel event. Around 80% of students leave home to attend university, moving an average 90 miles[2]. This synchronised event will increase population mixing at a national scale with the potential to spark outbreaks in new areas if not carefully managed. Once the university term starts there are more unique challenges facing universities. Students, in common with other 20- to 30-year olds, report high numbers of social contacts in their everyday lives[3]. Student accommodation frequently involves communal living, either in halls of residence that house several hundred students, or in all-student households renting in the private sector. Regular face-to-face teaching can involve several hundred students in a lecture theatre, and even without large lectures, tutorials and small group teaching involve close and prolonged contact between individuals.

The potentially high rate of transmission within a university setting is unlikely to translate to high morbidity among students. There is a marked age disparity in severe COVID-19 cases, with younger people less likely to exhibit typical symptoms or suffer severe outcomes[4]. In the UK at the time of writing, less than 0.2% of COVID-19-related deaths are in persons under 30. Students are typically young adults in their early twenties. Nevertheless, young adults are susceptible to infection and infectious to others. Hence, there is a risk of transmission within the student population, posing a risk to vulnerable students, people outside the university setting and family members when students return home.

A number of studies have investigated the challenges inherent in reopening of universities amidst the COVID-19 pandemic[5–9]. Existing models have mainly focused on isolated campus universities in the US, rather than civic universities that are common in the UK and elsewhere[6,9], and the majority have not had access to realistic mixing patterns within the university setting, which drive transmission. In this paper we combined analysis of social contact data with a data-driven mathematical modelling approach to investigate the impact of re-opening a UK university on COVID-19 transmission. We characterise patterns of disease transmission and investigate potential mitigating effects of interventions. These results are used to synthesise guidance on measures that universities might wish to consider for effective outbreak control once students arrive or return for the forthcoming academic year.

## Results

The Social Contact Survey (SCS) included 363 participants whose listed occupation included 'STUDENT'. Students reported more home contacts than other participants (3.5 versus 2.3, p value < 0.001). However, although students reported more contacts than other participants on average, there was no evidence of a systematic difference (29.9 versus 26.8, p value 0.40). The majority (82%, 95% CI: 79%, 86%) of students' social contacts are either home or associated with university. On average, students reported 20.0 (95% CI: 14.1, 28.8) university contacts, and 4.3 (95% CI: 2.7, 6.5) other/leisure contacts.

To capture student contact patterns within a university, we used comprehensive anonymised student accommodation data for the academic year 2019/2020 from the University of Bristol (UoB). The data included 20,819 registered undergraduates and 8501 registered postgraduates divided into 6 faculties and 28 schools and 2862 unique postcodes (see Supplementary Table 1 for number of students by year of study and faculty). Most students (92%) are under 30 years of age and the largest school is the School of Economics, Finance and Management with 3674 students.

We used the student data to create synthetic contact matrices for mixing between year-groups and schools. From postcodes we generated between school household contact matrices for each year of study, and for all years (Supplementary Notes 1 and 2 and Fig. 1). Halls of residence dominate the first-year contact matrix, with mixing across all schools and no clear assortative mixing (Fig. 1a).

In years 2 and 3, the average household size decreases substantially and there is increased assortativity mixing between schools, indicating that students are more likely to share accommodation with someone from their own school by choice (Fig. 1b, c).

The university-wide contact matrix consists of 161 groups of students categorised by 28 schools and nine year-groups (0, 1, 2, 3, 4, 5, 6, PGT (taught postgraduates), PGR (research postgraduates)) (Fig. 1d). The higher level of mixing between first years is evident in the lower left-hand corner and the assortative mixing by year and school is shown by the diagonal. There are fewer inter-year household contacts and more intra-university mixing between taught postgraduates than for research postgraduates.

We investigated the dynamics of an epidemic in the student population using a stochastic compartmental model with plausible COVID-19 parameter values (Table 1 and Fig. 2). Because of the population structure, the stochasticity and relatively small numbers involved, there is large intrinsic variability between simulations with identical parameter values; we report the mean and 98% prediction interval (PI).

Using plausible parameters (asymptomatic cases half as infectious as symptomatic cases and a reproduction number of $R_U = 2.7$, see Supplementary Note 3), and without interventions or holidays, we predict a university-wide outbreak with an early growth rate of 0.07 (98% PI: 0.03–0.10), which is equivalent to a doubling time of 9 days (98% PI: 7–24 days) (Fig. 3a). Based on the timescales of COVID-19 with baseline parameters, we expect that it would take around 4 months for the outbreak to peak, assuming no winter break.

First-year students drive the early part of the outbreak and experience the highest burden of infection, followed by second and third years and taught postgraduate students (Fig. 3b). Students in year 4 and above and research postgraduates have the lowest infection rates.

By the end of the first term, under the baseline model 4200 (98% PI: 1800–9800) students, or 15% (98% PI: 6–35%), have been infected. On the last day of term 54 (98% PI: 15–140) symptomatic cases and 640 (98% PI: 200–1600) asymptomatic cases are still infectious. On average, there are between 13 and 15 asymptomatic cases for every one symptomatic case. The number of cases doubles every 7–22 days. Without additional control measures, 68% (98% PI: 44–83%) of students would be infected by the end of the academic year. The low rate of symptoms and

**Fig. 1 Student mixing matrices based on shared accommodation.** The average number of students in each school sharing accommodation in **a** year 1, **b** year 2, **c** year 3 and **d** for all years and schools. The years are six undergraduate years: 0, 1, 2, 3, 4, 5 and two postgraduate groups R (research) and T (taught). The columns are ordered by total number of accommodation contacts. Data relate to the University of Bristol for the 2019/2020 academic year.

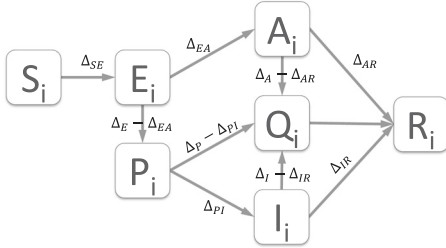

**Fig. 2 Model schematic.** Model flow diagram with infection states and rates between them for the stochastic meta-population model. The compartments are S: susceptible to infection, E: exposed, or infected but pre-infectious, P: pre-symptomatic and infectious, I: symptomatic and infectious, A: asymptomatic and infectious, Q: in quarantine, R: recovered and immune. The subscript $i$ refers to the subgroup. An explanation of the rates is given in the main text, Eqs. (1) and (2) and Table 1.

low morbidity rate results in a median of zero deaths in the student population.

The relative infectiousness of asymptomatic cases is central for determining the scale of a university-based outbreak. In our framework, asymptomatic cases are either less or as infectious as symptomatic cases; however, because asymptomatic cases do not self-isolate without a test, for higher values of relative infectiousness, $\varepsilon$, asymptomatic cases produce on average more secondary cases than symptomatic cases (see Supplementary Note 3 and Supplementary Fig. S3). For lower values of $\varepsilon$, university-focussed outbreaks are largely driven by the forcing from outside the university. For intermediate values, outbreaks peak after the first term. For high values, outbreaks peak before the end of the first term (see Supplementary Note 4 and Supplementary Figs. S4 and S5).

As a comparison to the baseline case, if asymptomatic cases are 30% as infectious as symptomatic cases ($R_U = 2.25$) then we expect an early growth rate of 0.06 (98% PI: 0.04–0.09) and a doubling time of 12 days (98% PI: 8–17 days). Without additional control measures, 36% (98% PI: 12–57%) of students would be infected by the end of the academic year. The epidemic profiles for the full 98% PI of potential scenarios for asymptomatic infectiousness, which corresponds to reproduction numbers from 1.7 to 3.4, are shown in Supplementary Fig. S5.

We investigated multiple interventions that reduced the infection burden in the student population (Fig. 3c–f). The impact of implementing each intervention was explored in isolation and in combination with other measures. When layering

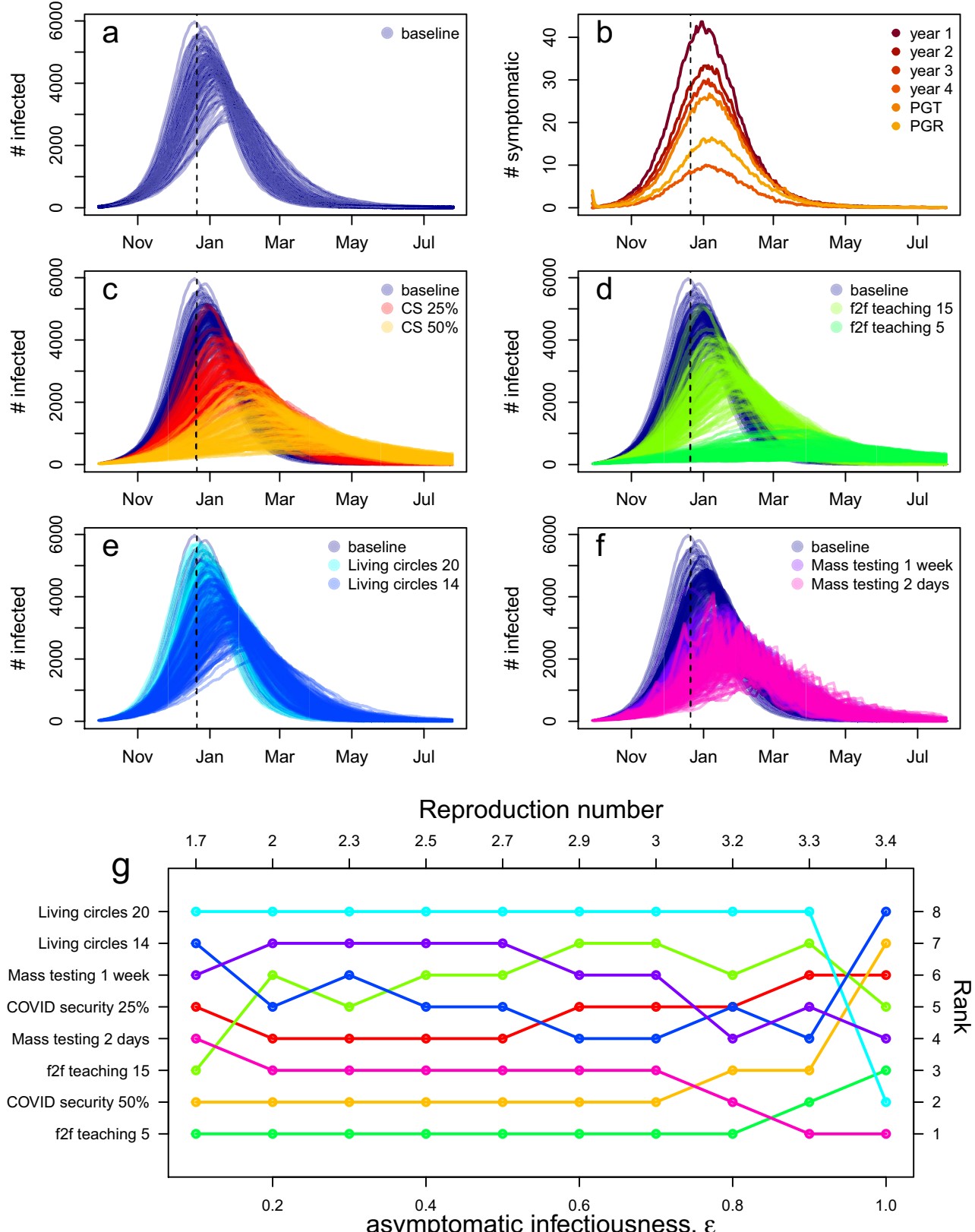

interventions, we implemented lower cost interventions first, such as creating COVID-secure interactions with face coverings and social/physical distancing, and reserved mass testing of non-symptomatic students as a more resource-intensive intervention.

For realistic values of COVID security and $R_U = 2.7$, we find that reducing the transmission probability with COVID-secure

interactions has the potential to reduce, but not completely eliminate, the size of outbreaks (Fig. 3c). We estimate that by reducing transmission for non-household contacts by 25% the early doubling time is between 7 and 20 days. The percentage of students infected by the end of the first term is 9% (98% PI: 4–28%) and the number of symptomatic and asymptomatic

**Fig. 3 Epidemic trajectories from the stochastic model. a** Epidemic trajectories for the total number of infected cases (symptomatic and asymptomatic cases) the baseline model from 100 realisations with best estimate parameters. **b** Mean number of symptomatic cases by year group from 100 realisations. Undergraduate years 1, 2, 3 and 4, taught postgraduates (PGT) and research postgraduates (PGR) are shown. **c** Epidemic trajectories when COVID security (CS) measures reduce transmission by 50 and 25%. **d** Epidemic trajectories when face-to-face teaching (f2f) is limited to 15 and 5 persons. **e** Epidemic trajectories for reduced living circles to 20 and 14 persons. **f** Epidemic trajectories when reactive mass testing is implemented every week and every 2 days. Dotted vertical lines denote the end of the first term. **g** Ranking of interventions by mean number of symptomatic cases at the end of the first term from 100 realisations for increasing values of asymptomatic infectiousness, and therefore also increasing values of the reproduction number. The colours correspond to the colours of the epidemic trajectories above.

students infectious on the last day of term is decreased to 30 (98% PI: 11–120) and 360 (98% PI: 120–1300), respectively. Reducing transmission for non-household contacts by 50% increases the doubling time to 13 (98% PI: 7–36) days and further reduces the number of infectious students on the last day of term 16 (98% PI: 2–57) symptomatic cases and 188 (98% PI: 59–650) asymptomatic cases.

Reducing the number of interactions made during face-to-face teaching from 20 to 15 other students increases the mean early doubling time to 11 (98% PI: 7–20) days and reduces the number of infected students at the end of the first term to 33 (98% PI: 10–110) symptomatic cases and 403 (98% PI: 120–1200) asymptomatic cases (Fig. 3d). Reducing the number of face-to-face contacts from 20 to 5 other students was the single most impactful intervention investigated in terms of the number of students infected by the end of the first term and the number of infectious students on the last day of term, increasing doubling time to 16 (98% PI: 9–110) days, including scenarios in which the number of cases in the student population was driven to zero (Fig. 3d). The number of infected students at the end of the first term was 11 (98% PI: 2–32) symptomatic cases and 140 (98% PI: 54–330) asymptomatic cases.

Implemented without other measures, reducing the size of living circles (defined as the number of students that share bathroom/kitchen facilities) from 24 to 20 or 14 students was overall the least effective intervention investigated (Fig. 3e and Supplementary Table 2). However, when implemented on top of COVID-secure interactions and reductions in face-to-face teaching, reducing living circles to 14 individuals does reduce the total percentage of students that are infected by the end of the first term by 25%.

Mass testing all students regardless of symptoms was effective at reducing the total number of infections and the initial rate of epidemic growth rate, but reactive testing was required for the whole year (Fig. 3f). Compared to other interventions, mass testing was generally more effective for higher values of the reproduction number and resulted in the third lowest number of infected students by the end of the first term. However, for lower values of asymptomatic infectiousness, and hence lower values of the reproduction number, reducing face-to-face teaching, implementing COVID security and reduced living circles were more effective than testing all students (Fig. 3g).

Testing all students every 2 days primarily reduced the number of students with asymptomatic and pre-symptomatic infections, reducing the ratio of asymptomatic to symptomatic cases to 9:1–11:1. However, the reduction in infection from mass testing comes at a substantial cost in terms of the number of students self-isolating: under 2 day testing, at the height of the outbreak, 1300 (860–1500) students (4.5, 3–5%) were self-isolating compared to 520 (98% PI: 470–560) students (1.9, 1.7–2.0%) in the baseline scenario.

Testing all students monthly had a minimal impact compared to not testing at all, reducing the average percentage of students

infected during the outbreak by 1.3%. Increasing testing frequency to fortnightly, weekly or every 3 or 2 days was beneficial, and this was robust to parameter choice (Supplementary Fig. S5).

We found that implementing multiple, layered interventions was able to effectively control transmission in the student population (Fig. 4a–c). The remaining cases in students were largely due to importation of infection from outside the university setting: reducing the background rate of infection demonstrates that if imported infections could be managed then the number of infected students could be very low.

## Discussion

Our results suggest that, under normal circumstances, COVID-19 would spread readily in a university setting. Our data-driven approach reveals natural heterogeneities in student mixing patterns that can be exploited to enhance disease control. We find that controlling transmission is possible with combinations of social distancing, online teaching, self-isolation and potentially mass testing of students without symptoms.

Our findings highlight the importance of monitoring first-year students and halls of residence in particular. In our analysis, first-year students experienced the highest rates of infection and dominate the early part of the outbreak due to the high levels of mixing in halls of residence. Since we first conducted this work in May 2020, a cross-sectional serosurvey of students in five English universities found that first-year undergraduates were three times more likely to be seropositive than other year groups and that seropositivity was 49% in students living in halls of residence in universities with high SARS-CoV-2 infection rates[10]. Halls of residence have been identified as a risk factor for the transmission of other close-contact infections including meningitis[11], mumps[12], norovirus[13], respiratory illnesses[14] and gastroenteritis[15]. In practice, students in larger residences are allocated into shared flats or living circles, potentially limiting widespread transmission. Limiting transmission within residences is paramount for COVID-19 control in university settings.

Lessons about infection control in universities can be learnt from other diseases. Mass vaccination used for meningitis, mumps and rubella outbreaks was not an option for COVID-19 in early 2020. During a mumps outbreak in a university hall of residence, Kay et al. reported difficulty in identifying higher risk students[12]. Due to the high number of contacts and of students' contact networks inhabit, universities may wish to consider how they might facilitate the collation of data to expedite the contact-tracing process. Embedding positive health behaviours like hand washing and using face coverings will also contribute to minimising transmission opportunities. A randomised control trial of hand washing in university residences found that installing alcohol hand sanitiser in every room, bathroom and dining hall reduced respiratory illness in students by 20%[14].

Previous modelling work, based on universities in the United States, has focussed on the necessity of regularly testing all students[5]. While our findings are consistent that frequent testing

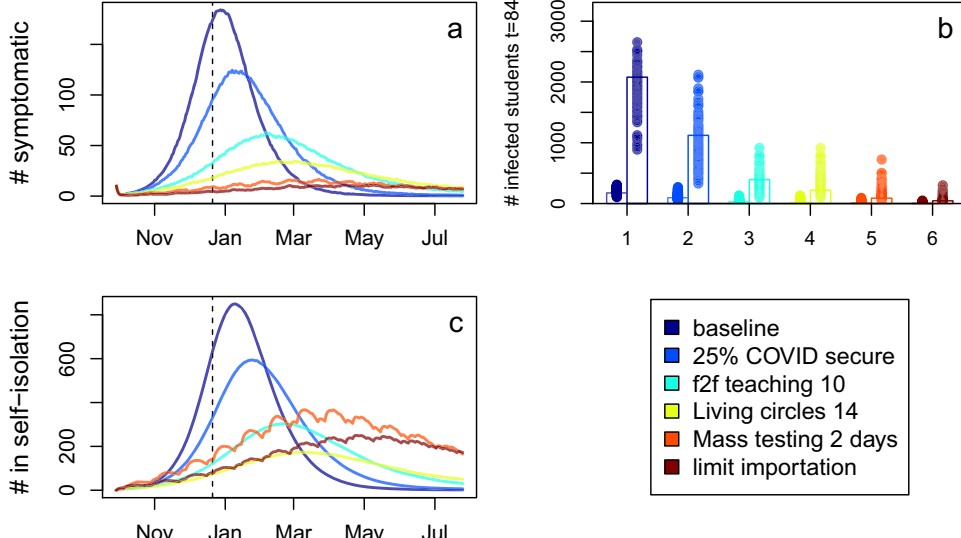

**Fig. 4 The impact of implementing layering interventions.** The intervention indicated in the legend is implemented in addition to the interventions above. **a** Number of symptomatic cases. **b** Number of infected (symptomatic (left bar) and asymptomatic (right bar)) students at the end of the first term (day 84). The height of the bar indicates the mean from 100 model replicates and the points show the individual model replicates. **c** Number of students that are self-isolating. The interventions considered are 25% COVID security (a 25% reduction in per contact transmission), limiting face-to-face (f2f) teaching to 10 persons, limiting living circles to 14 persons, mass asymptomatic testing every 2 days and limiting transmission from outside the university.

is necessary if used in isolation, our modelling approach demonstrates that other interventions are viable. This is partly due to our result that suggests that the reproduction number would be lower than previous studies have assumed due to the high proportion of asymptomatic cases. Furthermore, as previous studies have discussed[5], when prevalence is low, high testing burden can lead to unnecessary isolation of negative cases. We tried to mitigate test burden by implementing reactive mass testing once incidence increased. Antibody testing could also play in role in determining prior infection and infection rates in student populations.

Our work uses a similar compartmental modelling approach to the handful of models that have been developed for COVID-19 transmission in universities in the USA. A drawback of this approach is that individual behaviour is not readily captured; in particular, our approach does not capture superspreaders and contact tracing and isolation of contacts or living circles is difficult to include in detail. We capture some heterogeneity using household and faculty mixing data, and a stochastic model was necessary due to the potentially small number of students in each subgroup. Nevertheless, a network modelling approach would be more appropriate for studying superspreading events and individual-level variation.

Furthermore, while we had detailed data pertaining to the university student population, we had limited data on contact with the location population and we did not include university staff explicitly in the model. Given the age distribution of students, and the high likelihood of asymptomatic infection, staff and surrounding communities are likely to experience higher levels of morbidity than the students themselves. Although by-and-large students fraternise with students, they do pose some risk to more vulnerable groups within the university such as staff with co-morbidities, or to their local community. Reliable data on staff student interactions were not available and should be included in a future study. Safeguarding all is a high priority.

The aim of this work was to characterise potential COVID-19 transmission patterns in a university setting and identify strategies that may prove more likely to control transmission. This work was provided to the Scientific Pandemic Influenza Group on Modelling that provides modelling evidence to the UK

government[16] and used directly to inform planning for re-opening at the University of Bristol. In the absence of university outbreak data, we used COVID-19 transmission parameters estimated from other settings. Once the university year starts, and should there be an outbreak, this type of modelling should be used to estimate parameters in real time and provide a more accurate tool for guiding interventions.

## Methods
The SCS was a paper-based and online survey of 5388 unique participants in Great Britain conducted in 2010[3,17]. We have previously used these data to estimate the reproduction number for COVID-19[18]. The SCS included 363 participants whose listed occupation included 'STUDENT'. We extracted these participants to summarise their contacts by context (home, university, leisure/other, travel) and to estimate the potential COVID-19 reproduction number in students. We used Student's t-test to determine the level of evidence for the observed differences in numbers of contacts between students and the general population.

We used the SCS to estimate the contact rate between students by year and school. For a student in school/year group $i$, we assumed that all study contacts were within the same group, i.e., $s_{ii} = \bar{s}$ and $s_{ij} = 0$, where $\bar{s}$ is the mean number of university-associated contacts reported by students in the SCS. We assumed that non-study contacts occurred across the whole university. We took $r_{ij} = \bar{r}n_j/N$, where $\bar{r}$ is the mean number of other/leisure contacts reported by students in the SCS and $n_j/N$ is the proportion of students in group $j$.

We were provided with an anonymised extract of student data for a university relating to the 2019/2020 academic year. The study complied with the university data-protection policy for research studies (http://www.bristol.ac.uk/media-library/sites/secretary/documents/information-governance/data-protection-policy.pdf).

The data contained age, primary faculty affiliation (7 faculties), primary school affiliation (28 schools), year of study (6 undergraduate years, taught postgraduates and research postgraduates), term-time residence, home region (if in the UK) and country of origin for students registered in 2019/2020.

We used the university data to group students by school affiliation and year group—a total of 161 categories. We then estimate the household contact rate between students in each school/year group. We estimated the number of household contacts from the student data, taking postcode as a proxy for household. The average number of students in school/year group $j$ sharing accommodation with a student in group $i$ is calculated as follows:

$$h_{ij} = \frac{\sum_{k=1}^{P} n_{ik} n_{jk}}{\sum_{k=1}^{P} n_{ik}}$$

where $n_{ik}$ is the number of students in school/year $i$ living at unique postcode $k$ and $P$ is the number of unique postcodes.

In UoB, students in university residences will be assigned to a living circle, which is a group of students who have higher rates of contact. We take the baseline living circle size as 24 students and investigate the impact of smaller living circles. Where

the number of students at a single postcode exceeded the living circle size, we create subunits within the postcode that are the size of the living circle. Each living circle contains a random sample of students at that address. See Supplementary Fig. S1 for a pictorial explanation of how the data are processed.

We use a stochastic compartmental model to simulate transmission dynamics in the student population at UoB. We assumed that COVID-19 could be captured by seven infection states: susceptible to infection (S), latently infected (E), asymptomatic and infectious (A), pre-symptomatic and infectious (P), symptomatic and infectious (I), self-isolating (Q), hospitalised (H) and recovered and immune (R) with the subscript $i$ indicating the subgroup. The total number of students is given by $N$. The flow between compartments is depicted in Fig. 1 and given by the Eq. (1) below.

$$S_i(t + 1) = S_i(t) - \Delta_{SE}$$
$$E_i(t + 1) = E_i(t) + \Delta_{SE} - \Delta_E$$
$$A_i(t + 1) = A_i(t) + \Delta_{EA} - \Delta_A$$
$$P_i(t + 1) = P_i(t) + \Delta_E - \Delta_{EA} - \Delta_P \qquad (1)$$
$$I_i(t + 1) = I_i(t) + \Delta_{PI} - \Delta_I$$
$$R_i(t + 1) = R_i(t) + \Delta_{AR} + \Delta_{IR} + \Delta_{QR}$$
$$Q_i(t + 1) = Q_i(t) + \Delta_A - \Delta_{AR} + \Delta_I - \Delta_{IR} + \Delta_P - \Delta_{PI} - \Delta_{QR}$$

The transitions are given by:

$$\Delta_{SE} \sim Bin\left(S_i, 1 - \exp(-b - \sum_{j=1}^{n} \beta_{ij}(I_j + P_j + \varepsilon A_j + \delta_{ij}\varepsilon_Q Q_j)/N_j)\right)$$
$$\Delta_E \sim Bin(E_i, 1 - \exp(\sigma))$$
$$\Delta_{EA} \sim Bin(\Delta_E, 1 - \exp(-f))$$
$$\Delta_A \sim Bin(A_i, 1 - \exp(-\gamma_A - \tau_A))$$
$$\Delta_{AR} \sim Bin(\Delta_A, 1 - \exp(-\gamma_A/(\gamma_A + \tau_A))) \qquad (2)$$
$$\Delta_P \sim Bin(P_i, 1 - \exp(-\gamma_P - \tau_A))$$
$$\Delta_{PI} \sim Bin(\Delta_P, 1 - \exp(-\gamma_P/(\gamma_P + \tau_A)))$$
$$\Delta_I \sim Bin(I_i, 1 - \exp(-\gamma_I - \tau_I))$$
$$\Delta_{IR} \sim Bin(\Delta_I, 1 - \exp(-\gamma_I/(\gamma_I + \tau_I)))$$
$$\Delta_{QR} \sim Bin(Q_i, 1 - \exp(-\gamma_Q))$$

All state and transition variables are time dependent, although we have dropped ($t$) for convenience.

The student population was divided into 161 groups representing school and year of study. The proportion of students in each group and the mixing between groups was taken from the mixing matrix in Fig. 2. As 92% of the student population is under 30 years of age, we expect a high proportion of cases to be asymptomatic[19,20] ($f = 0.75$). In a survey of university students, a large range of symptoms were reported in participants who tested positive for COVID-19, suggesting that young adults may have symptoms, but not typical ones[21]. We also expect a low hospitalisation rate[22] ($h = 0.002$) and a low mortality rate of hospitalised cases ($\mu = 0.038$)[22] and have therefore not modelled hospitalisation and death here.

For symptomatic cases, we assume an average incubation period, during which cases are assumed not to be infectious and cannot be detected by the test, of $\langle 1/\sigma \rangle = 3$ days[23], after which they become infectious but pre-symptomatic for a period of $\langle 1/\gamma_P \rangle = 2$ days, when cases can be detected with a test. The infectious

period is taken as $\langle 1/\gamma \rangle = 3$ days[23], although there is uncertainty around these values. Symptomatic individuals are tested and moved to self-isolation at a rate $\tau_I$.

Asymptomatic cases are infectious for $\langle 1/\gamma_A \rangle = 5$ days, so that their average infectious period equals the infectious period for symptomatic cases. If asymptomatic cases tested, they are moved to self-isolation at rate $\tau_A$ where they remain for an average of 14 days. Individuals in self-isolation contribute to the force-of-infection within their subgroup only at a reduced rate $\varepsilon_Q = 0.5$.

The infectiousness of asymptomatic cases relative to symptomatic cases is represented by the parameter $\varepsilon$. It is accepted that asymptomatic transmission can and does occur; however, its relative importance has been difficult to measure due to consistent case definitions, incomplete sampling and follow-up[24]. Apparent asymptomatic transmission is often re-classified as pre-symptomatic transmission, i.e., transmission that occurs in the days before symptom onset[24–26]. The household secondary attack rate for truly asymptomatic index cases has been found to be lower than for symptomatic index cases[24,26–30]. A systematic review of 45 studies found that the secondary attack rate for asymptomatic index cases was 60% lower than for symptomatic index cases[29]. A different analysis of serological data from Switzerland found that asymptomatic cases had a 70% lower odds (34–88%) of infecting another household member compared to cases with symptoms[30]. Another modelling study of contact tracing estimated that transmission due to truly asymptomatic cases was limited, with pre-symptomatic and symptomatic transmission contributing the remainder in approximately equal proportions[31].

In summary, it appears that truly asymptomatic cases are less infectious than symptomatic cases. In order to capture the reduced infectiousness of asymptomatic cases, while acknowledging that students may report non-typical system, we take a baseline value for the relative infectiousness of asymptomatic cases, $\varepsilon$, of 0.5 and also consider a value of 0.3, and the full range of values in the Supplementary Note 4. We assume that pre-symptomatic and symptomatic cases are equally infectious[32].

We assume the transmission rate between group $i$ and group $j$, $\beta_{ij}$, is proportional to the contact rate $c_{ij}$, where $c_{ij}$ is the average number of contacts in group $j$ made by a person in group $i$. We assume that contacts were either household contacts ($h_{ij}$), study contacts ($s_{ij}$) or random contacts ($r_{ij}$), so each entry in the contact matrix is given by $c_{ij} = h_{ij} + s_{ij} + r_{ij}$. In this formulation, we assume an equal probability of transmission by contact type. In order to translate the contact matrix into the transmission matrix, we calculate a constant $k$ such that the maximum eigenvalue of the transmission matrix B $= \{\beta_{ij}\} = \{kc_{ij}\}$ equals the reproduction number[33]. There is an additional background rate of infection, governed by the parameter $b$.

To estimate the reproduction number in the student population, we took a population-wide reproduction number of $R_0 = 2.7$, calculated as follows. In our framework, if a symptomatic case generates $R_s$ secondary cases, then an asymptomatic case will generate $R_A = \varepsilon R_S$ secondary cases. With $R_0 = R_S + R_A$, $R_s = R_0/(f + (1 - f)\varepsilon)$. If cases without symptoms are 50% as infectious as cases with symptoms ($\varepsilon = 0.5$), and a fraction $f = 0.6$ of the general population has symptoms when infected, then in a university setting when a lower proportion of cases have symptoms ($f = 0.25$) but have on average 10% more contacts than an average person, we would expect a reproduction number of $R_U = 2.7$. If $\varepsilon = 0.1$ then $R_U = 1.7$; if $\varepsilon = 1$ then $R_U = 3.4$ (see Supplementary Note 3 and Supplementary Fig. S3).

For the initial conditions, we assumed that 0.2% of incoming students had active asymptomatic or pre-symptomatic infections, which we assigned randomly across the year/faculty groups. For each scenario, we ran the model 100 times using a different random seed. The model was simulated for 1 year to illustrate the full range of dynamics, and we consider the state of the outbreak after 84 days, which is

## Table 1 Baseline model parameter values, meaning and sources.

| Parameter | Symbol | Value/Range | References |
|---|---|---|---|
| Number of household contacts between subgroups | $h_{ij}$ | Estimated from accommodation data | |
| Number of study contacts between subgroups | $s_{ij}$ | 20.0 (SD: 4.0) | 3,17 |
| Number of university-wide contacts between subgroups | $r_{ij}$ | 4.3 (SD: 1.0) | 3,17 |
| Basic reproduction number in the UK | $R_0$ | 2.7 | 18,38 |
| Transmission probability per contact per day | $\beta$ | Estimated from reproduction number | |
| Proportion of cases with no symptoms | $f$ | 0.75 | 19,20 |
| Average infectious period | $1/\gamma$ | 3 days | 23,35 |
| Average incubation period | $1/\sigma$ | 3 days | 23,35 |
| Average pre-symptomatic period | $1/\gamma_P$ | 2 days | 23,35 |
| Average infectious period for asymptomatic case | $1/\gamma_A$ | $1/\gamma_P + 1/\gamma$ | |
| Average time to test for symptomatic cases | $1/\tau_I$ | 2 days | 35 |
| Average time to test for asymptomatic cases | $1/\tau_A$ | Asymptomatic cases not tested in baseline model | |
| Length of time in self-isolation | $1/\gamma_Q$ | 14 days | |
| Relative infectiousness of asymptomatic cases compared to symptomatic cases | $\varepsilon$ | 0.5 (0.3–0.7) | 24,26–30 |
| Reduction in infectiousness whilst in self-isolation | $\varepsilon_Q$ | 0.5 | Assumption |
| Background rate of infection | $b$ | – | Assumption |

the number of days between the start of the September term and the winter holidays at the end of the first term. For numerical results, we report the mean and 98% PI from 100 simulations. The model code is available at https://github.com/ellen-is/unimodel[34].

For the main results reported in the paper, we ran the model with baseline parameters for 100 realisations, drawing from distributions indicated in Table 1 and report the 98% PI calculated directly from the 1st and 100th order statistic. To understand the variability further, we run 100 further realisations of the model varying all baseline parameters independently by ±10%, and report those results in the Supplementary information.

The impact of the infectiousness of asymptomatic cases was explored for values of $\varepsilon$ between 0 (asymptomatic cases not infectious) and 1 (asymptomatic cases as infectious as symptomatic cases), which corresponds to reproduction numbers ranging from 1.7 to 3.4 (see Supplementary Note 2).

We assumed that symptomatic cases would be tested and self-isolate within 48 h, which is consistent with the median time between symptom onset and test of 2 days in the UK[35]. Contact tracing is difficult to implement explicitly in the compartmental model framework, but the mechanism of action can be captured by a lower within-group transmission rate. We focussed on interventions that could be implemented on top of wider control measures and were guided on feasibility by the UoB Scientific Advisory Group. We considered the following pragmatic interventions, limited by feasibility (see Table 2 for a summary):

- Baseline conditions are 'business as usual' behaviour within universities with PHE guidelines. Symptomatic cases are tested are moved into self-isolation after an average of 48 h if test positive. No additional testing for people with no symptoms. Students are assumed to be in living circles that comprise of a maximum of 24 individuals to reflect existing UoB arrangements.
- COVID security represents the reduction in transmission associated with social distancing and the use of face coverings[36,37]. We modelled COVID security by reducing the transmission probability associated with non-household contacts by 25 and 50% to capture the impact of face covering use and social distancing outside of residences.
- Reduced face-to-face teaching is captured by reducing the number of face-to-face teaching contacts by 25 and 75% from 20 students to 15 and then 5 students.
- Reduced living circles reflects reducing the number of students sharing facilities within accommodation. In the baseline scenario, we assumed that students were in contact with other students living in the same accommodation, forming household groups up to a maximum of 24 individuals. For accommodation with more than 24 residents, we divided the accommodation population up into subunit 'living circles' of 24 students. To explore the impact of living circle size, we reduced the maximum living circle size from 24 to 20 and then 14 persons.
- Reactive mass testing: we simulate scenarios in which all students are tested for the presence of current infection if the number of test-positive cases in a given week is greater than the previous week. If mass testing is triggered in the model, all students are tested within a given number of days, which is varied between 2 and 7 days. Additional testing is continued until the number of test-positive cases in a given week is less than the previous week.
- Multiple, layered interventions: we investigated the impact of each of the above interventions in isolation and then applied sequentially: 25% reduction in transmission due to COVID security, followed by a reduction in face-to-face teaching to ten study contacts, followed by a reduction in living circles to 24 individuals, and reactive mass testing every 2 days if the infection rate on campus should rise, and finally a reduction in importation rates from outside the university population.

For each model realisation we calculated (a) the doubling time during the exponential growth phase as $\ln(2)/r$, where $r$ is the exponential growth rate in the number of infected individuals, (b) the incident number of symptomatic and asymptomatic cases at the end of the first term (day 84 of the model), (c) the time the outbreak turns over, (d) the number of students in self-isolation and (e) the ratio of asymptomatic to symptomatic cases.

We ranked the interventions when implemented without additional measures by mean number of symptomatic cases at the end of the first term calculated from 100 realisations of the model for a given set of parameters and repeated this ranking for values of $\varepsilon$ between 0 (asymptomatic cases not infectious) and 1 (asymptomatic cases as infectious as symptomatic cases).

**Reporting summary**. Further information on research design is available in the Nature Research Reporting Summary linked to this article.

## Data availability
The Social Contact Survey data used in this study are available at http://wrap.warwick.ac.uk/54273/. The raw UoB student data are protected and are not available due to data privacy laws. The aggregated UoB student contact matrices are available to download at https://github.com/ellen-is/unimodel/[34].

**Table 2 Intervention scenarios for controlling transmission within university settings.**

| Scenario | Transmission probability per contact household/other per day | Mean no. of random contacts (SD = 1) | Mean no. of within-course contacts (SD = 4) | Max living circle size | % transmission reduction due to self-isolation within/between groups | Asymptomatic testing |
|---|---|---|---|---|---|---|
| Baseline | 0.05[a] | 4[b] | 20[b] | 24 | 50/100 | None |
| COVID security | 0.05[a]/0.04 or 0.025 | 4[b] | 20[b] | 24 | 50/100 | None |
| Reduced face-to-face teaching | 0.05[a]/0.05[a] | 4[b] | 15 or 5 | 24 | 50/100 | None |
| Reduced living circle size | 0.05[a]/0.05[a] | 4[b] | 20[b] | 20 or 14 | 50/100 | None |
| Improved self-isolation | 0.05[a]/0.05[a] | 4[b] | 20[b] | 24 | 100/100 | None |
| Reactive mass testing | 0.05[a]/0.05[a] | 4[b] | 20[b] | 24 | 50/100 | Every 2 or 7 days when rates are increasing |
| Multiple | 0.05[a]/0.04 | 4[b] | 5 | 14 | 50/100 | Every 2 days when rates are increasing |

[a]Calculated such that $R = 2.7$.
[b]Estimated from the Social Contact Survey.

## Code availability

The model was coded using R version 4.0.02 (2020-06-22). Model code for reproducing epidemic trajectories and other outputs is available at https://github.com/ellen-is/unimodel/[34].

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

## Acknowledgements

Thanks to TJ McKinley for advice in developing the code. Thanks also to the three reviewers whose comments improved the manuscript. We thank the University of Bristol for providing the data and the University of Bristol Scientific Advisory Group for support. E. B.-P. and L. D. are supported by Medical Research Council (MRC) (MC/PC/19067), and L. D. is supported by EPSRC EP/V051555/1 and The Alan Turing Institute EPSRC EP/N510129/1. E. B.-P., H. C., E. N. and L. D. are funded via the JUNIPER Consortium (MR/V038613/1). A. C. T. is funded by the Wellcome Trust (217509/Z/19/Z) and the MRC (MR/V028545/1). E. B.-P., H. C., K. T. and M. H. acknowledge support from the NIHR Health Protection Research Unit in Behavioural Science and Evaluation at the University of Bristol. The Health Protection Research Unit (HPRU) in Behavioural Science and Evaluation at University of Bristol is part of the National Institute for Health Research (NIHR) and a partnership between University of Bristol and Public Health England (PHE), in collaboration with the MRC Biostatistics Unit at University of Cambridge and University of the West of England. We are a multidisciplinary team undertaking applied research on the development and evaluation of interventions to protect the public's health. Our aim is to support PHE in delivering its objectives and functions. Follow us on Twitter: @HPRU_BSE.

## Author contributions

E.B.-P. conceived the study, developed and ran the model and wrote the first draft of the paper. L.D. developed the data processing code. E.B.-P., H.C., L.D. and C.R. designed the study. E.B.-P., H.C., L.D., A.T., A.C.T., G.H., K.T., A.F., M.H., E.N. and C.R. interpreted the results. E.B.-P., H.C., L.D., A.T., A.C.T., G.H., K.T., A.F., M.H., E.N. and C.R. wrote and revised the paper.

## Competing interests

The authors declare no competing interests.
