## [Peer Review File · Nature Communications]

REVIEWER COMMENTS

Reviewer #1 (Remarks to the Author):

This is an important piece of work that I imagine has already had a large impact on university plans over the summer. While it has some limitations around parameter uncertainty and the ability to capture tracing and individual behaviour (which the authors acknowledge) these should not be impediments to publication. It would not be possible or reasonable for this work to include considerations of the current student outbreaks that will have occurred after submission of this work, and again I do not think this should be an impediment to publication so long as it is clear when the work was completed and submitted for publication.

Overall, I judge this to be important work completed in a time of considerable uncertainty making use of reasonable modelling choices.

Comments and questions:

Within the description of how the SCS data is used on Page 2: could the authors clarify what is an i -indexed group here - is it within a year, school pair or is it year or school? E.g.: would a first-year physics student be allocated \overline{s} contacts with other first-year physics students, or with other first-year or with other physics? I believe it is with other first-year physics students, but wanted to clarify. I think this is clarified later in the 'Model parameters' section, but consider adding something earlier.

Am I right in understanding that in the use of the SCS data, study contacts are (as I asked above) entirely within-year/program group, whereas non-study contacts are uniformly random over the student population? If so, the authors may wish to add a sentence summarising this intuition.

Could the authors clarify why the use of a mean number of contacts from the SCS, rather than a distribution of some kind? I imagine it's just for simplicity and parsimony, but it would be good to hear if that's true.

In using the Bristol data extract, the authors note that 'Where the number of students at a single postcode exceeded the living circle size, we create subunits within the postcode that are the size of the living circle.' - is this only for students in university residences? If so, could the authors comment on the possible impact of this assumption in the context of their Bristol-specific data? For example, I believe that in my block of flats at the same postcode we have approximately 18 students living across 5 different households, but this may be unusual at Bristol.

In the description of the modelling framework: The compartmental diagram in Figure 1 includes a 'D' compartment, but I believe this is not described in the text or equations in the 'Modelling framework' section - admittedly deaths are unlikely in the student population, but could the authors consider aligning the figure and the text?

A few notational queries in the description of the modelling framework:

- Are the i subscripts on the compartments meant to indicate a group? If so, could the authors add a clarifying phrase?
- Is t used for both time and the parameters t_I and t_A ? This is a little confusing - consider using a different notation for the flow parameters (unless this is entirely standard, in which case feel free to refute and ignore!)
- I believe that all the δ values are time-dependent, and I believe that using them in this way with no time-dependence in their sub or superscript is fairly standard. However, would the authors consider

adding a phrase mentioning this time dependence?

- I notice that in the dynamical expressions all compartments are time-dependent (e.g. $P_i(t)$), but in the delta expressions this is not always the case - for example in the expression for Δ_{SE} we have $S_i(t)$ but just P_j without the indication of being a function of time (I see similar throughout the delta-expressions). If this is a minor notation error, could the authors correct it, or if it is intentional could they add some explanatory text?

In the "Model parameters" section I am confused about what is depicted in Figure 2, and how this interacts with the use of the SCS and the Bristol data extract. I had thought that the mixing matrix would be a combination of the accommodation data from the Bristol data and the study and non-study contacts, and this seems to be confirmed on page 5, but the caption for Figure 2 suggests that the matrices shown there are based only on shared accommodation, and the first few sentences of the 'Model parameters' section suggests that these are the basis of the mixing process. Could the authors clarify what is depicted in Figure 2, and whether it is the entire mixing matrix, or just the h_{ij} component?

In Figure 2 there are no a,b,c,d labels - could the authors add them?

Am I correct in understanding that in the baseline transmission model contacts within and outwith the household are equally likely to transmit? If so, this is likely an important assumption and the authors may wish to add some additional discussion of this. While it is unfair to ask the authors to include information on the current university outbreaks (taking place after the submission of this manuscript) it is worth considering that, for example, the large number of early cases in student accommodation at the University of Glasgow were detected before teaching started, face-to-face or otherwise. I am not suggesting that the authors discuss this or other examples, but instead that a discussion of the household/non-household transmission probability might help future researchers to re-use this work most appropriately in the context of new information on current outbreaks.

The authors say that "We estimated the number of infected students at the start of term using home location and incidence in home location as of July 2020" - at the regional level from the ONS survey and via overseas information, or by some other means? Could the authors consider clarifying this point? Apologies if this is in the SI and I have missed it.

I was pleased to see that the code is available online on GitHub, and that is licensed in a way that will allow extension and re-use. If the authors are inclined, they might want to include a commit reference or a persistent code object (e.g. using zenodo) just to ensure that the version of the code used for this paper will remain available even if the repo is updated. Upon inspection of the code it appears well-written if not especially engineered within the main model code. It is reasonably simple to run and understand, and could be easily re-implemented and verified if needed. The data files included are a little hard to understand, so if the authors feel it would be appropriate within their data rules they may wish to include some documentation.

A small point of clarification on Table 1 - apologies if this is stated and I've just missed it which is quite likely! - are these transmission probabilities per day?

I applaud the authors for publishing their findings on student mixing in addition to their overall results on mitigations. These mixing findings are a valuable resource for other modellers, and I appreciate the efforts to do this within the bounds of data protection.

In Figure 3b), I notice an initial decline in the number of symptomatic individuals - is this just a result

of the seeding procedure?

In the results reported in Figure 4 - am I correct in understanding that the order of layering interventions is downwards in the legend - that is, baseline, then just 25% covid secure, then 25% covid secure+ f2f teaching 5, then 25% covid secure+ f2f teaching 5 + Living circles 14, etc? Please consider adding clarifying text in the figure caption.

Typos/very minor suggestions:

In Methods: Social Contact Survey data there is a leftover '[ref]'

In figure 4 consider a different capitalisation for 'covid' in 'covid secure'

I imagine that it is unimportant, but the 'Software policy' PDF does not display properly for me even with the suggested Adobe software, so I have not reviewed it.

(-- Jess Enright)

Reviewer #2 (Remarks to the Author):

This paper addresses an important issue and the modeling appears to have been well done. But I have questions about how the sensitivity analyses were done and how uncertainty in results is communicated.

Regarding the sensitivity analyses, the description is very vague. I understand that parameter values are varied, but were they varied individually or jointly? The former is quite restrictive. And it is not clear where/how sensitivity results are reported? It does not help that there is a note in the sensitivity analysis section that indicates: see SI section 2. But there is not mention of any sections in the paper. Perhaps the authors meant Figure 2?

Intervals are provided in the Results section but these are never described except for some 95% CI at the start. Does these intervals represent ranges from the sensitivity analyses? The issue of uncertainty needs to be much more fully addressed for results to be interpretable.

Finally, how does/should the model allow for the superspreader phenomena that has often been observed? Explicit discussion of how limits of modeling might impact interpretation of results would be important to include in Discussion.

Reviewer #3 (Remarks to the Author):

This study used a stochastic compartmental model to investigate COVID-19 transmission and control measures using detailed data on students at the University of Bristol, which the authors use to generalize to other civic university settings. The authors find that without countermeasures, a large proportion of the study body would become infected, and that no single countermeasure halts transmission; however, the proportion who become infected could be significantly reduced by layering multiple countermeasures.

MAJOR COMMENTS

It is often necessary to rely on assumptions rather than data in modeling studies. For example, here the authors appear to assume that 100% of symptomatic individuals will be tested within 48 hours, and that a "COVID-secure" policy can reduce overall transmission by 50%. The limitations introduced by relying on such assumptions can be partially overcome by sensitivity analyses, as the authors, to their credit, include here. In other cases, the assumptions are relatively unimportant and a single-best-estimate, even if not entirely accurate, is sufficient. However, when too many assumptions become layered on top of each other, sensitivity analyses can effectively end up saying that anything could happen. That appears to be the case here, as shown in Fig. S4, indicating that at the peak of infections, between 0.2 and 60 individuals will be symptomatic.

Thus, my central concern with this study is that it is based too heavily on assumptions that are not sufficiently informed by data: especially the relative infectiousness of asymptomatic individuals, R_0 , and the relative transmission rates (per person and per hour) in classroom versus household settings. The conclusions of the study are sensitive to the first of these, as shown by the sensitivity analysis, and would likely be highly sensitive to the other two as well. One solution to this snowballing uncertainty, as noted by the authors in the Discussion, is to calibrate the model to actual epidemiological data: it should be quickly apparent if asymptomatic infectiousness is closer to 0.1 or 1.0, for example; or whether the secondary infections of an index case are primarily classroom or household contacts. In the absence of such data to constrain such important model parameters, it is reasonable to ask whether modeling can really tell us much at this stage.

Similarly, the intervention scenarios struck me as somewhat arbitrary: why is COVID security modeled as 25% and 50%? As an alternative, while still imperfect, the authors could have cited an actual study of mask efficacy (e.g. <https://www.sciencedirect.com/science/article/pii/S0140673620311429>) and then combined this with observed or assumed mask compliance. Why is the testing delay assumed to be 48 hours, rather than the 24 hours that is already common in many places, or 5 hours as the University of Illinois has been able to achieve (<https://covid19.illinois.edu/health-and-support/on-campus-covid-19-testing/>)? Why were face-to-face teaching scenarios represented by reducing 20 contacts to 5, rather than 10? Why were living circle scenarios represented by reducing 24 contacts to 20 or 14, and not 5? In the absence of data or other constraints, it seems that an "apples to apples" comparison would be most appropriate -- e.g., reduce each by the same amount, say 50%. Lastly, one may ask: do the scenarios reveal something that could not be learned simply by taking the assumed R_0 value and multiplying it by the intervention effectiveness? One does not need a dynamic model to know that if R_0 is 1.7 and you apply an intervention that reduces transmission by 50%, the epidemic will be controlled.

My remaining major concerns are laid out clearly by the authors themselves in the Discussion. The most significant of these is the exclusion of staff, who are a significant proportion of the university population, and who are at much higher risk of serious complications due to their age distribution. It also does not appear that the extra-university community was included in the model (despite being mentioned in the abstract). Mixing between students and the community is a major issue even for campus universities, but is even more so for civic universities. Given typical student living situations, part-time jobs, leisure time, etc., it seems likely that a university outbreak would quickly spread into the community (and, conversely, that a community outbreak would quickly seed infections in the university). Thus, modeling the student population in isolation does not seem to adequately capture real-world transmission dynamics.

Overall, while this study is well written has much to commend it -- in particular, the detailed contact network data -- its central conclusions do not appear to be sufficiently robust to either (a) uncertainty in the model parameters, including asymptomatic transmissibility, R_0 , and intervention effectiveness; or (b) the omission of staff and the community from the transmission dynamics.

MINOR COMMENTS

1. p. 1, Abstract: In "reduced face-to-face testing", presumably "teaching" was meant.

2. p. 2, Introduction: The authors mention the low expected mortality rate among students; however, no mention is made of university staff, who comprise a large proportion of the total number of people at the university (e.g. for Oxford, roughly 14,000 staff to 24,000 students; source: <https://www.ox.ac.uk/about/facts-and-figures/staff-numbers>). While not all staff are old enough to be at high COVID risk, they are nonetheless important to include in the analysis (as are, potentially, the 8% of students who are over the age of 30).

3. p. 2, "Nevertheless, it appears likely that young adults are susceptible to infection and infectious to others." -- The hedging here is a bit odd; there is no longer any doubt about this (see e.g. <https://www.sciencedirect.com/science/article/abs/pii/S0163445320301171>).

4. p. 2, "Hence there is a risk of asymptomatic transmission within the student population, posing a risk to vulnerable students, people outside the university setting and family members when students return home." -- The omission of university staff is again noted. It is not clear why only asymptomatic transmission is mentioned here; part of the reason for the spike of infections at the University of Illinois was from students failing to isolate who were not merely symptomatic but even diagnosed (<https://www.nature.com/articles/d41586-020-02611-y>).

5. p. 2, Introduction: I am concerned that only two citations are given for studies investigating COVID transmission in universities, including a self-citation to a literature review conducted in early July. While it is true that keeping up with the COVID literature is itself a full-time job, and that COVID literature reviews become outdated virtually as soon as they are published, three months is still a long period to go with no updates, especially since most of the work on COVID transmission in universities has been published in this period. Aside from the University of Illinois example given above, which received significant press coverage, other examples include <https://www.medrxiv.org/content/10.1101/2020.07.06.20147272v1>, <https://arxiv.org/abs/2008.09597>, and <https://github.com/bu-racs/BU-COVID> (I would guess there are many others in addition to these). There is also of course a large body of literature on COVID modeling for schools, which of course share some similarities with universities (e.g. low-risk student vs. high-risk staff populations, in-person vs. remote learning, etc.).

6. p. 2, Methods: I assume "[ref]" is supposed to be a reference.

7. p. 3, "We estimated the number of household contacts from the student data, taking postcode as a proxy for household.": My (admittedly not very detailed) understanding of the British postcode system is that typically at most a single building would be assigned a postcode; i.e., an apartment block would be assigned a single postcode. Assuming that everyone living in that apartment block is in the same "living circle" would seem to very significantly overestimate the number of contacts, as it would treat connections the same for people within an apartment versus in different apartments.

8. p. 3, Methods: To model detailed networks of individual students, it seems odd to me to use a

compartmental model rather than an agent-based model. This would seem to introduce a well-mixed assumption into the model that would not be reflected in either an agent-based model or in reality.

9. p. 4, Methods: Perhaps I am missing it, but I am unable to find the definitions of all the terms of the equations, e.g. "b".

10. p. 4, Methods: 3 days is an extremely short infectious period; although this might capture the period of highest infectiousness, it surely does not represent the whole infectious period. Some citations are Nishiura et al., who estimate significant infectiousness out to ~15 days (<https://www.sciencedirect.com/science/article/pii/S1201971220301193>), and Wolfel et al., who describe high viral loads for "the first week of symptoms" (<https://www.nature.com/articles/s41586-020-2196-x>).

11. p. 5, Methods: The assumption of $R_0 = 2.7$ would seem to be central to the analysis. This is problematic for two reasons: (1) It is not clear why this number was chosen (e.g. what was the citation? E.g. <https://www.sciencedirect.com/science/article/pii/S2590053620300410> or <https://www.koreascience.or.kr/article/JAKO202016151586067.page>). (2) As those two papers show, there is considerable variability and/or uncertainty in its value, so a sensitivity analysis is essential, in addition to the sensitivity in asymptomatic infectiousness already considered.

12. p. 5, Initial conditions: Why was the model run for a year if only 84 days are considered for the results? The longer the model is run, the more problematic the exclusion of the broader community and its dynamics becomes.

13. p. 5, Initial conditions: The authors are commended for releasing their code. I had to make a few changes to get it to run however (replacing "isnan" with "std::isnan" in the C++ component, and renaming the file from "betamat.RData" to "betamat.Rdata").

14. p. 5, Sensitivity analysis: It would be preferable to include actual uncertainty estimates (e.g., in the R_0 estimate in the citations listed above) rather than simply multiplying each parameter by +/- 10%, which is certainly a significant underestimate for some parameters (e.g. infectious period).

15. p. 5, Control options: It is indeed more difficult to model contact tracing in compartmental models, but it can be done, and it would seem important to do so: see Sturniolo et al., <https://www.medrxiv.org/content/10.1101/2020.05.14.20101808v2>. It seems highly optimistic to assume that 100% of symptomatic cases are tested and fully isolated within 48 hours; to my knowledge this has not been achieved in practice anywhere in the world, certainly not in the UK.

16. p. 6, Control options: The description of "reactive mass testing" is not clear to me. Every single student is tested? What does it mean "within 2 to 28 days"?

17. p. 6, "where r is the exponential growth rate in the number of infected individuals, the incident number of symptomatic and asymptomatic cases at the end of the first term (day 84 of the model), the time the outbreak turns over, the number of students in self-isolation and ratio of asymptomatic to symptomatic cases" -- I am a little confused by this sentence.

18. p. 7, Contact patterns: 3.5 home contacts seems far fewer than the "living circles" of 24. In any case, shouldn't the distribution of numbers of home contacts in the model be drawn from the distribution of home contacts in the SCS data? (Possibly another reason to use an agent-based model.)

19. p. 7, "Although students do not have more contacts than the general population, 18 to 24-year

olds do have more contacts than the wider population." -- I am not sure what to make of this. It would seem to imply that only 18 to 24-year-olds who are not students have more contacts than the wider population, which would seemingly not be relevant here. Likewise, "Taken in combination with the contact duration" is mentioned without prior mention of what the contact durations are or how they are used.

20. p. 7, Typo: "assortativit".

21. p. 7, Transmission dynamics: It is not appropriate to report the minimum and maximum results since these depend on the number of samples taken (e.g., the minimum and maximum values from a normal distribution with 100 samples might be ± 2.8 ; with 10,000 samples they are more like ± 3.6). The 95% confidence interval should be reported instead, preferably based on a larger number of simulations (e.g. 1000) to improve repeatability.

22. p. 8, Transmission dynamics: Why are there more than 10 times as many asymptomatic vs. symptomatic cases (880 vs. 73) on the last day? The methods (p. 4) state 75% asymptomatic. I assume this is counting presymptomatic and asymptomatic together, and is skewed due to the assumption of the relatively long presymptomatic infectious period (2 days) and the extremely short symptomatic infectious period (3 days). In addition, Viner et al. seems to be the wrong reference for 75% asymptomatic in children, as this review does not discuss symptomatic rates; <https://arxiv.org/pdf/2006.08471.pdf> might be a better source.

23. p. 9, Transmission dynamics: It surprises me that "reducing the size of the living circles" was the least effective intervention, given high rates of household transmission, although this is perhaps just an artifact of the intervention being smaller magnitude (i.e. 24 to 14 contacts, vs. 20 to 5 contacts for teaching). Was the rate of transmission per contact the same for a class-based context versus a living-based context? Epidemiological evidence suggests that household transmission rates are much higher, due to the greater time spent in close proximity (and perhaps e.g. the absence of good ventilation/HVAC systems in households relative to universities and workplaces).

24. p. 9, Discussion: The statement "Maintaining social distancing between living circles within residences is paramount for maintaining COVID-19 control" would seem to be at odds with the earlier statement "reducing the size of living circles ... was overall the least effective intervention investigated". The former would seem to align better with the realities encountered at universities that have reopened, such as the University of Illinois mentioned above.

25. p. 10, Discussion: A lot of conclusions seem to hinge on asymptomatic cases having lower transmissibility. However, evidence suggests this is not the case, e.g. <https://www.sciencedirect.com/science/article/pii/S1201971220302502>. While I am aware that this was included in the sensitivity analysis, statements like "the reproduction number would be lower than previous studies have assumed due to the high proportion of asymptomatic cases" seems to itself be an assumption, not a finding.

26. p. 10, "when prevalence is low the false positive rate can exceed the true positive rate leading to unnecessary isolation of negative cases" -- PCR tests have close to zero false positive rates; only antibody tests have high false positive rates (see <https://doi.org/10.1515/dx-2020-0091>). The study cited (Paltiel et al.) was a hypothetical modeling study not using data from actual COVID PCR tests.

27. p. 12, Fig. 2: If the schools were sorted by size, the figures might be easier to interpret. In addition, while interesting, it is not clear how the separation into schools/faculties really make a

difference in the analysis.

-- Cliff Kerr

REVIEWER COMMENTS

Reviewer #1 (Remarks to the Author):

This is an important piece of work that I imagine has already had a large impact on university plans over the summer. While it has some limitations around parameter uncertainty and the ability to capture tracing and individual behaviour (which the authors acknowledge) these should not be impediments to publication. It would not be possible or reasonable for this work to include considerations of the current student outbreaks that will have occurred after submission of this work, and again I do not think this should be an impediment to publication so long as it is clear when the work was completed and submitted for publication.

Overall, I judge this to be important work completed in a time of considerable uncertainty making use of reasonable modelling choices.

Thank you for the positive and useful comments.

Comments and questions:

Within the description of how the SCS data is used on Page 2: could the authors clarify what is an i-indexed group here - is it within a year, school pair or is it year or school? E.g.: would a first-year physics student be allocated \overline{s} contacts with other first-year physics students, or with other first-year or with other physics? I believe it is with other first-year physics students, but wanted to clarify. I think this is clarified later in the 'Model parameters' section, but consider adding something earlier.

That's right – the groups are school and year. We've added a sentence to clarify. "We used the university data to group students by school affiliation and year group – a total of 161 categories."

Am I right in understanding that in the use of the SCS data, study contacts are (as I asked above) entirely within-year/program group, whereas non-study contacts are uniformly random over the student population? If so, the authors may wish to add a sentence summarising this intuition.

Thanks, this was communicated mathematically, but we have now added a description:

"We used the SCS to estimate the contact rate between students by year and school. For a student in school/year group i , we assumed that all study contacts were within the same group, i.e. $s_{ij} = \bar{s}$ and $s_{ij} = 0$, where \bar{s} is the mean number of university-associated contacts reported by students in the SCS. We assumed that non-study contacts occurred across the whole university. We took $r_{ij} = \bar{r} n_j / N$, where \bar{r} is the mean number of other/leisure contacts reported by students in the SCS and n_j / N is the proportion of students in group j ."

Could the authors clarify why the use of a mean number of contacts from the SCS, rather than a distribution of some kind? I imagine it's just for simplicity and parsimony, but it would be good to hear if that's true.

We originally used the mean value to illustrate the intrinsic variability (which is quite large), however we have reproduced the results by draw from distributions derived from the data.

In using the Bristol data extract, the authors note that 'Where the number of students at a single postcode exceeded the living circle size, we create subunits within the postcode that are the size of the living circle.' - is this only for students in university residences? If so, could the authors comment on the possible impact of this assumption in the context of their Bristol-specific data? For example, I

believe that in my block of flats at the same postcode we have approximately 18 students living across 5 different households, but this may be unusual at Bristol.

We applied this approach to all postcodes, not just university residences, so in the situation described we would create multiple living circles for the block of flats.

In the description of the modelling framework: The compartmental diagram in Figure 1 includes a 'D' compartment, but I believe this is not described in the text or equations in the 'Modelling framework' section - admittedly deaths are unlikely in the student population, but could the authors consider aligning the figure and the text?

Thanks for pointing this out. We've removed the D category from the figure.

A few notational queries in the description of the modelling framework:

-Are the β subscripts on the compartments meant to indicate a group? If so, could the authors add a clarifying phrase?

We have added additional explanation shortly before the equations are given.

- Is β used for both time and the parameters β_I and β_A ? This is a little confusing - consider using a different notation for the flow parameters (unless this is entirely standard, in which case feel free to refute and ignore!)

We have changed t_I to τ_I and t_A to τ_A

- I believe that all the delta values are time-dependent, and I believe that using them in this way with no time-dependence in their sub or superscript is fairly standard. However, would the authors consider adding a phrase mentioning this time dependence?

We have added a statement clarifying that all transition variables are time dependent.

- I notice that in the dynamical expressions all compartments are time-dependent (e.g. $P_i(t)$), but in the delta expressions this is not always the case - for example in the expression for Δ_{SE} we have $S_i(t)$ but just P_j without the indication of being a function of time (I see similar throughout the delta-expressions). If this is a minor notation error, could the authors correct it, or if it is intentional could they add some explanatory text?

We have removed the additional (t)s in the transition equations.

In the "Model parameters" section I am confused about what is depicted in Figure 2, and how this interacts with the use of the SCS and the Bristol data extract. I had thought that the mixing matrix would be a combination of the accommodation data from the Bristol data and the study and non-study contacts, and this seems to be confirmed on page 5, but the caption for Figure 2 suggests that the matrices shown there are based only on shared accommodation, and the first few sentences of the 'Model parameters' section suggests that these are the basis of the mixing process. Could the authors clarify what is depicted in Figure 2, and whether it is the entire mixing matrix, or just the h_{ij} component?

Figure 2 is only the household mixing, h_{ij} . We have clarified that in the results section that refers to figure 2.

In Figure 2 there are no a,b,c,d labels - could the authors add them?

Added.

Am I correct in understanding that in the baseline transmission model contacts within and outwith the household are equally likely to transmit? If so, this is likely an important assumption and the authors may wish to add some additional discussion of this. While it is unfair to ask the authors to include information on the current university outbreaks (taking place after the submission of this manuscript) it is worth considering that, for example, the large number of early cases in student accommodation at the University of Glasgow were detected before teaching started, face-to-face or otherwise. I am not suggesting that the authors discuss this or other examples, but instead that a discussion of the household/non-household transmission probability might help future researchers to re-use this work most appropriately in the context of new information on current outbreaks.

This is a good point. It is correct that we assumed an equal probability of transmission for household, study and other contacts, which is probably not the case. We used this assumption because we didn't have data to guide anything different, but we have added this assumption to the methods. This assumption could be improved in future model instantiations. We have added a clarifying sentence to the methods.

"In this formulation, we assume an equal probability of transmission by contact type."

The authors say that "We estimated the number of infected students at the start of term using home location and incidence in home location as of July 2020" - at the regional level from the ONS survey and via overseas information, or by some other means? Could the authors consider clarifying this point? Apologies if this is in the SI and I have missed it.

Apologies – this was a detail we didn't include. We have now simplified the initial conditions as it has no bearing on the results by assuming that 0.2% of incoming students are infected at the start of term.

I was pleased to see that the code is available online on GitHub, and that is licensed in a way that will allow extension and re-use. If the authors are inclined, they might want to include a commit reference or a persistent code object (e.g. using zenodo) just to ensure that the version of the code used for this paper will remain available even if the repo is updated. Upon inspection of the code it appears well-written if not especially engineered within the main model code. It is reasonably simple to run and understand, and could be easily re-implemented and verified if needed. The data files included are a little hard to understand, so if the authors feel it would be appropriate within their data rules they may wish to include some documentation.

Thanks for taking the time to look at the code and thanks for this suggestion. We are keeping the github version up-to-date with the changes from this revision, and as soon as the manuscript is finalised and accepted we will create a persistent code object using zenodo as suggested.

A small point of clarification on Table 1 - apologies if this is stated and I've just missed it which is quite likely! - are these transmission probabilities per day?

We have added "per day" to Table 1 – thanks.

I applaud the authors for publishing their findings on student mixing in addition to their overall results on mitigations. These mixing findings are a valuable resource for other modellers, and I appreciate the efforts to do this within the bounds of data protection.

In Figure 3b), I notice an initial decline in the number of symptomatic individuals - is this just a result of the seeding procedure?

The reviewer is correct, because we didn't initially seed the "correct" proportions of asymptomatic and symptomatic cases. We have now assumed that all incoming cases are asymptomatic or presymptomatic, as we feel this is more realistic, although there is still a small decline at the start.

In the results reported in Figure 4 - am I correct in understanding that the order of layering interventions is downwards in the legend - that is, baseline, then just 25% covid secure, then 25% covid secure+ f2f teaching 5, then 25% covid secure+ f2f teaching 5 + Living circles 14, etc? Please consider adding clarifying text in the figure caption.

Thanks, we have added text to the caption.

Typos/very minor suggestions:

In Methods: Social Contact Survey data there is a leftover '[ref]'

Thanks – now removed.

In figure 4 consider a different capitalisation for 'covid' in 'covid secure'. I imagine that it is unimportant, but the 'Software policy' PDF does not display properly for me even with the suggested Adobe software, so I have not reviewed it.

We refer to the editor for guidance on this.

(-- Jess Enright)

Reviewer #2 (Remarks to the Author):

This paper addresses an important issue and the modeling appears to have been well done. But I have questions about how the sensitivity analyses were done and how uncertainty in results is communicated.

Regarding the sensitivity analyses, the description is very vague. I understand that parameter values are varied, but were they varied individually or jointly? The former is quite restrictive. And it is not clear where/how sensitivity results are reported? It does not help that there is a note in the sensitivity analysis section that indicates: see SI section 2. But there is not mention of any sections in the paper. Perhaps the authors meant Figure 2?

Thanks for the comment. We have updated our approach to parameter uncertainty based on this and the other reviewers' comments. We have clarified in the methods that the main results relate to the baseline parameters with contact numbers drawn from distributions and we have added a table containing the baseline parameters. We varied the parameter independently – and clarified this in the main text. The sections are in the supplementary information (SI), rather than the main paper.

“For the main results reported in the paper, we ran the model with baseline parameters for 100 realisations, drawing from distributions where indicated in Table 1. To understand the variability further, we run 100 further realisations of the model varying all baseline parameters independently by +/-10%, and report those results in the Supplementary Information (SI).”

Intervals are provided in the Results section but these are never described except for some 95% CI at the start. Does these intervals represent ranges from the sensitivity analyses? The issue of uncertainty needs to be much more fully addressed for results to be interpretable.

The intervals we report now are 98% prediction intervals and they come from the baseline parameter values. We had described in the Results section – we have added this to the methods and clarified when we've reported the prediction interval as opposed to other intervals.

Finally, how does/should the model allow for the superspreader phenomena that has often been

observed? Explicit discussion of how limits of modeling might impact interpretation of results would be important to include in Discussion.

There are reports of superspreader events, but we do not capture them in this formulation. We mention in the discussion that network models would be suited to capture superspreader events as our approach assumes homogeneity between individuals. We have elaborated on this in the discussion.

Reviewer #3 (Remarks to the Author):

This study used a stochastic compartmental model to investigate COVID-19 transmission and control measures using detailed data on students at the University of Bristol, which the authors use to generalize to other civic university settings. The authors find that without countermeasures, a large proportion of the study body would become infected, and that no single countermeasure halts transmission; however, the proportion who become infected could be significantly reduced by layering multiple countermeasures.

MAJOR COMMENTS

It is often necessary to rely on assumptions rather than data in modeling studies. For example, here the authors appear to assume that 100% of symptomatic individuals will be tested within 48 hours, and that a "COVID-secure" policy can reduce overall transmission by 50%.

We accept the general point that we needed to provide more justification for the intervention scenarios. We address each point below, including the testing window and the reduction associated with COVID-security.

The limitations introduced by relying on such assumptions can be partially overcome by sensitivity analyses, as the authors, to their credit, include here. In other cases, the assumptions are relatively unimportant and a single-best-estimate, even if not entirely accurate, is sufficient. However, when too many assumptions become layered on top of each other, sensitivity analyses can effectively end up saying that anything could happen. That appears to be the case here, as shown in Fig. S4, indicating that at the peak of infections, between 0.2 and 60 individuals will be symptomatic.

We considered carefully how to approach the challenge of unknown parameters, and we disagree that including a full parameter exploration means that nothing can be concluded.

Firstly, we intentionally included the full range of simulations for all values of asymptomatic infectiousness in the supplementary information in order to highlight the importance of asymptomatic transmission.

Secondly, while absolute peak infection burden is dependent on asymptomatic infectiousness, we demonstrated that the choice of intervention is robust to assumptions about infectiousness. Figure 3D shows that irrespective of assumptions about asymptomatic transmission, the ranking of interventions is robust. That means that policy decisions about interventions can be made without improved estimates of asymptomatic transmission. We highlight that the absolute numbers of cases from the model are not predictions, and should be used to guide understanding, rather than for the precise numbers.

Thus, my central concern with this study is that it is based too heavily on assumptions that are not sufficiently informed by data: especially the relative infectiousness of asymptomatic individuals, R_0 , and the relative transmission rates (per person and per hour) in classroom versus household settings. The conclusions of the study are sensitive to the first of these, as shown by the sensitivity analysis, and would likely be highly sensitive to the other two as well.

We have explored the impact of relative infectiousness and R0 but not transmission by setting. Figure S2 in the supplementary information demonstrates that by varying asymptomatic infectiousness, we are also varying R0 – therefore the sensitivity analysis improves insight into the impact of R0 on the system.

As also noted by reviewer 1, we assumed a single transmission rate for transmission in multiple settings. This is a simplifying assumption which we have clarified in the main paper. It is beyond the scope of this study to estimate the relative transmission rates in different settings, especially in the absence of data on transmission by setting.

Nevertheless, we agree that providing some guidance to the readers is useful. Therefore, we have added a paragraph to the methods which discusses the potential values of the asymptomatic infectiousness parameter in more detail, referencing systematic reviews. We have also removed the results for equal infectiousness from the abstract and the main paper, as we now believe this scenario is implausible.

One solution to this snowballing uncertainty, as noted by the authors in the Discussion, is to calibrate the model to actual epidemiological data: it should be quickly apparent if asymptomatic infectiousness is closer to 0.1 or 1.0, for example; or whether the secondary infections of an index case are primarily classroom or household contacts. In the absence of such data to constrain such important model parameters, it is reasonable to ask whether modeling can really tell us much at this stage.

As argued above, we disagree that scenario modelling is not useful in advance of an outbreak and indeed a number of insights from this study have proven correct. For example, our analysis highlighted the importance of communal accommodation and the higher infection rates in first year students in advance of the university outbreaks. As mentioned above, we have narrowed down our plausible parameter ranges for asymptomatic infectiousness. One of the challenges in fitting to observed outbreak data is that the test behaviour varies enormously during outbreaks, particularly in this low symptomatic group, and this is largely unquantified.

Similarly, the intervention scenarios struck me as somewhat arbitrary: why is COVID security modeled as 25% and 50%? As an alternative, while still imperfect, the authors could have cited an actual study of mask efficacy (e.g. <https://www.sciencedirect.com/science/article/pii/S0140673620311429>) and then combined this with observed or assumed mask compliance.

Thanks for pointing out this study. It came out after the original analysis was performed, but we make use of it now. This study did not include studies from UK, however general conclusions can be drawn. The reported risk reduction in non-healthcare settings was smaller than for healthcare settings, and also smaller for single layer face masks as opposed to respirators and surgical masks. The reported relative risk in non-healthcare settings 0.56 (95% CI 0.4, 0.79), therefore which is consistent with our plausible estimate of a 50% reduction. Our values are comparable to other modelling studies and we also now refer to effectiveness measured in field studies.

Why is the testing delay assumed to be 48 hours, rather than the 24 hours that is already common in many places, or 5 hours as the University of Illinois has been able to achieve (<https://covid19.illinois.edu/health-and-support/on-campus-covid-19-testing/>)?

We have assumed an average time to being tested of 48 hours for symptomatic cases, which means in the model an exponential distribution with 50% of individuals tested within 48 hours. This value is reflective of the UK data, where the average time between symptom onset and test date is just over 2 days.

Why were face-to-face teaching scenarios represented by reducing 20 contacts to 5, rather than 10? Why were living circle scenarios represented by reducing 24 contacts to 20 or 14, and not 5? In the absence of data or other constraints, it seems that an "apples to apples" comparison would be most appropriate -- e.g., reduce each by the same amount, say 50%.

The interventions considered were guided by the university based on what was feasible. Some areas are easier to target than others, for example, it is possible to reduce face-to-face teaching by 100% but it is not possible to reduce living circles by 100%. Therefore, we decided that reducing face-to-face teaching contacts and living circles contacts by the same amount would be equivalent. We have noted this in the methods.

Lastly, one may ask: do the scenarios reveal something that could not be learned simply by taking the assumed R_0 value and multiplying it by the intervention effectiveness? One does not need a dynamic model to know that if R_0 is 1.7 and you apply an intervention that reduces transmission by 50%, the epidemic will be controlled.

The same results are not obtained by simply multiplying 1.7 by 0.5, as we do not apply COVID-security to household contacts. Thus, it is the relative magnitude of household contacts compared to study and random contacts, plus the structure of those contacts that determines the outbreak size. Another example is the impact of increased mixing of first years compared to order years.

My remaining major concerns are laid out clearly by the authors themselves in the Discussion. The most significant of these is the exclusion of staff, who are a significant proportion of the university population, and who are at much higher risk of serious complications due to their age distribution. It also does not appear that the extra-university community was included in the model (despite being mentioned in the abstract).

We decided not to include staff in this model as we didn't have the data on which to base staff-student contacts. We are currently running a contact survey to measure this and future versions of the model may include staff.

Mixing between students and the community is a major issue even for campus universities, but is even more so for civic universities. Given typical student living situations, part-time jobs, leisure time, etc., it seems likely that a university outbreak would quickly spread into the community (and, conversely, that a community outbreak would quickly seed infections in the university). Thus, modeling the student population in isolation does not seem to adequately capture real-world transmission dynamics.

Transmission between students and the wider community is not clear, especially while social distancing measures are in place. In the UK, a number of university outbreaks did not result in significant transmission to the wider population, so we argue that modelling the student population in isolation is justified.

Overall, while this study is well written has much to commend it -- in particular, the detailed contact network data -- its central conclusions do not appear to be sufficiently robust to either (a) uncertainty in the model parameters, including asymptomatic transmissibility, R_0 , and intervention effectiveness; or (b) the omission of staff and the community from the transmission dynamics.

MINOR COMMENTS

1. p. 1, Abstract: In "reduced face-to-face testing", presumably "teaching" was meant.

Thanks.

2. p. 2, Introduction: The authors mention the low expected mortality rate among students; however, no mention is made of university staff, who comprise a large proportion of the total number of people at the university (e.g. for Oxford, roughly 14,000 staff to 24,000 students; source: <https://www.ox.ac.uk/about/facts-and-figures/staff-numbers>). While not all staff are old enough to be at high COVID risk, they are nonetheless important to include in the analysis (as are, potentially, the 8% of students who are over the age of 30).

We have emphasised the lack of analysis of impact on staff as a limitation, which is due to the lack of reliable contact data for staff. This lack of data would require strong assumptions that we feel would detract from the main focus of the paper on student interactions driving the university epidemic.

We made the simplifying assumption of a homogeneous age population of students as this was not the focus of the study, and future work could look in more detail at subjects with older students.

3. p. 2, "Nevertheless, it appears likely that young adults are susceptible to infection and infectious to others." -- The hedging here is a bit odd; there is no longer any doubt about this (see e.g. <https://www.sciencedirect.com/science/article/abs/pii/S0163445320301171>).

We have removed the first part of this sentence.

4. p. 2, "Hence there is a risk of asymptomatic transmission within the student population, posing a risk to vulnerable students, people outside the university setting and family members when students return home." -- The omission of university staff is again noted. It is not clear why only asymptomatic transmission is mentioned here; part of the reason for the spike of infections at the University of Illinois was from students failing to isolate who were not merely symptomatic but even diagnosed (<https://www.nature.com/articles/d41586-020-02611-y>).

Thanks for pointing out this article. We have removed 'asymptomatic' from the sentence.

5. p. 2, Introduction: I am concerned that only two citations are given for studies investigating COVID transmission in universities, including a self-citation to a literature review conducted in early July. While it is true that keeping up with the COVID literature is itself a full-time job, and that COVID literature reviews become outdated virtually as soon as they are published, three months is still a long period to go with no updates, especially since most of the work on COVID transmission in universities has been published in this period. Aside from the University of Illinois example given above, which received significant press coverage, other examples include <https://www.medrxiv.org/content/10.1101/2020.07.06.20147272v1>, <https://arxiv.org/abs/2008.09597>, and <https://github.com/bu-racs/BU-COVID> (I would guess there are many others in addition to these). There is also of course a large body of literature on COVID modeling for schools, which of course share some similarities with universities (e.g. low-risk student vs. high-risk staff populations, in-person vs. remote learning, etc.).

We tried to reference appropriately and mainly referenced UK studies as they are the most relevant. We have added some additional pertinent references that have been published since the first submission.

6. p. 2, Methods: I assume "[ref]" is supposed to be a reference.

Fixed, thank you.

7. p. 3, "We estimated the number of household contacts from the student data, taking postcode as a proxy for household.": My (admittedly not very detailed) understanding of the British postcode

system is that typically at most a single building would be assigned a postcode; i.e., an apartment block would be assigned a single postcode. Assuming that everyone living in that apartment block is in the same "living circle" would seem to very significantly overestimate the number of contacts, as it would treat connections the same for people within an apartment versus in different apartments.

We don't assume that everyone in at a postcode is in the same living circle – that's why we have a maximum living circle size. If, say, 240 students are registered at the postcode we would divide them into 10 living circles. While this is an approximation, university accommodation (halls) is typically comprised of individual rooms with shared bathrooms and either shared kitchens or catering facilities as well as social areas. Such a set up facilitates contacts between students across the entire hall, as do social areas. Halls accommodation is mostly limited to first year students and students in higher years tend to live in private rented houses that are much more dispersed through the city. Our analysis of the distribution of student numbers across postcodes points towards much smaller households, that rarely share a postcode with another household.

8. p. 3, Methods: To model detailed networks of individual students, it seems odd to me to use a compartmental model rather than an agent-based model. This would seem to introduce a well-mixed assumption into the model that would not be reflected in either an agent-based model or in reality.

We chose to use a metapopulation framework which makes the most of the information we have and makes the simplifying modelling assumption that contacts that we don't have information on are made uniformly at random. Indeed, the differences in contact patterns between 1st year students and other years (Fig 1) are reflected in very different epidemic trajectories for those year groups. Using an agent-based model would introduce assumptions not supported by the data.

9. p. 4, Methods: Perhaps I am missing it, but I am unable to find the definitions of all the terms of the equations, e.g. "b".

Thanks, we have updated the methods section for clarity and completeness, and also added a table of parameters and definitions.

10. p. 4, Methods: 3 days is an extremely short infectious period; although this might capture the period of highest infectiousness, it surely does not represent the whole infectious period. Some citations are Nishiura et al., who estimate significant infectiousness out to ~15 days (<https://www.sciencedirect.com/science/article/pii/S1201971220301193>), and Wolfel et al., who describe high viral loads for "the first week of symptoms" (<https://www.nature.com/articles/s41586-020-2196-x>).

The average infectious period in the model is 5 days, which includes 2 days pre-symptomatic and 3 days with symptoms and is exponentially distributed, so it will capture infectious periods of 15 days and longer. The values we selected are consistent with our meta-analysis of the literature (Challen et al 2020).

11. p. 5, Methods: The assumption of $R_0 = 2.7$ would seem to be central to the analysis. This is problematic for two reasons: (1) It is not clear why this number was chosen (e.g. what was the citation?

E.g. <https://www.sciencedirect.com/science/article/pii/S2590053620300410> or <https://www.koreascience.or.kr/article/JAKO202016151586067.page>). (2) As those two papers show, there is considerable variability and/or uncertainty in its value, so a sensitivity analysis is essential, in addition to the sensitivity in asymptomatic infectiousness already considered.

One key point we make in the paper, that is not captured in other university modelling studies, is that the university reproduction number is affected by the high proportion of asymptomatic cases. We included a paragraph in the methods that explains where this number has come from as well as further explanation in the supplementary materials.

“To estimate the reproduction number in the student population, we took a population-wide reproduction number of $R_0 = 2.7$, calculated as follows. In our framework, if a symptomatic case generates R_S secondary cases, then an asymptomatic case will generate $R_A = \varepsilon R_S$ secondary cases. With $R_0 = R_S + R_A$, $R_S = R_0 / (f + (1 - f)\varepsilon)$. If cases without symptoms are 50% as infectious as cases with symptoms ($\varepsilon = 0.5$), and a fraction $f = 0.6$ of the general population has symptoms when infected, then in a university setting when a lower proportion of cases have symptoms ($f = 0.25$) but have on average 10% more contacts than an average person, we would expect a reproduction number within university of $R_U = 2.7$. If $\varepsilon = 0.1$ then $R_U = 1.7$; if $\varepsilon = 1$ then $R_U = 3.4$ (see SI, section 2, figure S2).”

12. p. 5, Initial conditions: Why was the model run for a year if only 84 days are considered for the results? The longer the model is run, the more problematic the exclusion of the broader community and its dynamics becomes.

84 days is the number of days until the end of the first academic term, but we also report on dynamics to the end of the academic year. It is appropriate to run the model for a year so that the difference between reproduction numbers can be fully understood. As mentioned above, it generally appears that university outbreaks in the UK have been fairly contained within university populations, so we do not believe this is a very poor assumption.

13. p. 5, Initial conditions: The authors are commended for releasing their code. I had to make a few changes to get it to run however (replacing "isnan" with "std::isnan" in the C++ component, and renaming the file from "betamat.RData" to "betamat.Rdata").

Thank you. We are pleased that the model could be tested relatively easily. We are continuing to update the code on GitHub.

14. p. 5, Sensitivity analysis: It would be preferable to include actual uncertainty estimates (e.g., in the R_0 estimate in the citations listed above) rather than simply multiplying each parameter by +/-10%, which is certainly a significant underestimate for some parameters (e.g. infectious period).

The reason we tested +/- 10% was to test the uncertainty in the model. The distributions that the reviewer is referring to are individual-level distributions, thus we think that it is not the right approach to use these as population-level values. For example, the individual-level impact of a small proportion of cases being infected for 15+ days is completely different to the impact of increasing the mean infectious period to over 15 days.

15. p. 5, Control options: It is indeed more difficult to model contact tracing in compartmental models, but it can be done, and it would seem important to do so: see Sturniolo et al., <https://www.medrxiv.org/content/10.1101/2020.05.14.20101808v2>.

We don't think that contract tracing is central to this model as there is limited traditional contact tracing in the university setting, however there are certainly refinements that can be made to the model in the future.

It seems highly optimistic to assume that 100% of symptomatic cases are tested and fully isolated within 48 hours; to my knowledge this has not been achieved in practice anywhere in the world, certainly not in the UK.

This is a misunderstanding of the model implementation - we have assumed an average time to being tested of 48 hours for symptomatic cases, which means in the model an exponential distribution with 50% of individuals tested within 48 hours.

16. p. 6, Control options: The description of "reactive mass testing" is not clear to me. Every single student is tested? What does it mean "within 2 to 28 days"?

Thanks. We have reworded this section.

“We simulate scenarios in which all students are tested for the presence of current infection if the number of test-positive cases in a given week is greater than the previous week. If mass testing is triggered in the model, all students are tested within a given number of days, which is varied between 2 and 7 days.”

17. p. 6, "where r is the exponential growth rate in the number of infected individuals, the incident number of symptomatic and asymptomatic cases at the end of the first term (day 84 of the model), the time the outbreak turns over, the number of students in self-isolation and ratio of asymptomatic to symptomatic cases" -- I am a little confused by this sentence.

Apologies, we are simply enumerating the observables from our model. It now reads:

“For each model realisation we calculated a) the doubling time during the exponential growth phase as $\ln(2)/r$, where r is the exponential growth rate in the number of infected individuals, b) the incident number of symptomatic and asymptomatic cases at the end of the first term (day 84 of the model), c) the time the outbreak turns over, d) the number of students in self-isolation and e) the ratio of asymptomatic to symptomatic cases.”

18. p. 7, Contact patterns: 3.5 home contacts seems far fewer than the "living circles" of 24. In any case, shouldn't the distribution of numbers of home contacts in the model be drawn from the distribution of home contacts in the SCS data? (Possibly another reason to use an agent-based model.)

The 3.5 home contacts numbers is from the Social Contact Survey. We don't use this value because the sample size is small, and we have comprehensive student accommodation data from UoB. However, an average household size of 3.5 is consistent with the student accommodation data with a maximum living circle size of 24 – in those data the mean living circle size is 4. Although an agent-based model is a potential approach, it also involves drawbacks as it requires many more assumptions about who mixes with whom.

19. p. 7, "Although students do not have more contacts than the general population, 18 to 24-year olds do have more contacts than the wider population." -- I am not sure what to make of this. It would seem to imply that only 18 to 24-year-olds who are not students have more contacts than the wider population, which would seemingly not be relevant here. Likewise, "Taken in combination with the contact duration" is mentioned without prior mention of what the contact durations are or how they are used.

We have removed these sentences because they are not necessary for the paper.

20. p. 7, Typo: "assortativit".

Corrected, thanks

21. p. 7, Transmission dynamics: It is not appropriate to report the minimum and maximum results since these depend on the number of samples taken (e.g., the minimum and maximum values from a normal distribution with 100 samples might be +/- 2.8; with 10,000 samples they are more like +/- 3.6). The 95% confidence interval should be reported instead, preferably based on a larger number of simulations (e.g. 1000) to improve repeatability.

Thanks. We have re-run all the simulations and now report the prediction intervals because we think it is more informative to provide an interval that will capture future observations for repeatability.

22. p. 8, Transmission dynamics: Why are there more than 10 times as many asymptomatic vs. symptomatic cases (880 vs. 73) on the last day? The methods (p. 4) state 75% asymptomatic. I assume this is counting presymptomatic and asymptomatic together, and is skewed due to the assumption of the relatively long presymptomatic infectious period (2 days) and the extremely short symptomatic infectious period (3 days).

There are more asymptomatic cases for two reasons: because there are asymptomatic and presymptomatic cases, and also because symptomatic cases are removed due to testing.

In addition, Viner et al. seems to be the wrong reference for 75% asymptomatic in children, as this review does not discuss symptomatic rates; <https://arxiv.org/pdf/2006.08471.pdf> might be a better source.

Thanks for this reference.

23. p. 9, Transmission dynamics: It surprises me that "reducing the size of the living circles" was the least effective intervention, given high rates of household transmission, although this is perhaps just an artifact of the intervention being smaller magnitude (i.e. 24 to 14 contacts, vs. 20 to 5 contacts for teaching). Was the rate of transmission per contact the same for a class-based context versus a living-based context? Epidemiological evidence suggests that household transmission rates are much higher, due to the greater time spent in close proximity (and perhaps e.g. the absence of good ventilation/HVAC systems in households relative to universities and workplaces).

Yes, this is an artifact of the relatively smaller reduction in living circle size. As discussed above, this was made for pragmatic reasons. It is correct that evidence suggests that household transmission is more likely than transmission elsewhere, however to our knowledge, there is no data related to student households which are highly likely to be very different to family households. Furthermore, as the data on contact patterns outside the household are limited, any adjustments to the transmission probability by setting would be arbitrary.

24. p. 9, Discussion: The statement "Maintaining social distancing between living circles within residences is paramount for maintaining COVID-19 control" would seem to be at odds with the earlier statement "reducing the size of living circles ... was overall the least effective intervention investigated". The former would seem to align better with the realities encountered at universities that have reopened, such as the University of Illinois mentioned above.

This is due to the fact that accommodation drives epidemics, but practical constraints mean that reducing living circle size sufficiently would not be implementable.

25. p. 10, Discussion: A lot of conclusions seem to hinge on asymptomatic cases having lower transmissibility. However, evidence suggests this is not the case, e.g. <https://www.sciencedirect.com/science/article/pii/S1201971220302502>. While I am aware that this was included in the sensitivity analysis, statements like "the reproduction number would be lower than previous studies have assumed due to the high proportion of asymptomatic cases" seems to itself be an assumption, not a finding.

Our results are robust for the case when asymptomatic cases are as infectious per contact as symptomatic cases. Because asymptomatic cases are tested less frequently than symptomatic cases, this could result in a higher reproduction number for asymptomatic cases. The new observation in the paper is that the reproduction number hinges on the relative infectiousness of asymptomatic cases. We have expanded the justification for the values considered for the asymptomatic transmission rate. The study referenced above does not contain enough detail to differentiate asymptomatic from pre-symptomatic cases and goes against evidence from systematic reviews, which suggest that asymptomatic cases are less infectious than symptomatic cases.

26. p. 10, "when prevalence is low the false positive rate can exceed the true positive rate leading to unnecessary isolation of negative cases" -- PCR tests have close to zero false positive rates; only antibody tests have high false positive rates (see <https://doi.org/10.1515/dx-2020-0091>). The study cited (Paltiel et al.) was a hypothetical modeling study not using data from actual COVID PCR tests.

There are multiple reasons why frequent testing is undesirable. We have edited the sentence in question.

"as previous studies have discussed[5], when prevalence is low, high testing burden can lead to unnecessary isolation of negative cases. We tried to mitigate test burden by implementing reactive mass testing once incidence increased"

27. p. 12, Fig. 2: If the schools were sorted by size, the figures might be easier to interpret. In addition, while interesting, it is not clear how the separation into schools/faculties really make a difference in the analysis.

We have received positive comments about these figures from policy makers and stakeholders. The mixing matrix is sorted by year group, highlighting the increased contacts among first years. Dividing the university population into schools and faculties is essential for limiting the speed of spread predicted by a random mixing model.

-- Cliff Kerr

REVIEWER COMMENTS

Reviewer #1 (Remarks to the Author):

I am content with the revisions made by the authors, and believe that (subject to the other reviewers being satisfied) this work is suitable for publication.

I am pleased that the authors are willing to generate a persistent zenodo link for the code commit that will accompany the final version of the manuscript, and cheerfully remind them to do so when the time comes. As a regular user of code from published manuscripts I can attest that knowing exactly the version used to produce the results in the manuscript is very valuable. Thank you!

Reviewer #2 (Remarks to the Author):

I believe that the authors have done an excellent job in responding to reviewer comments—particularly with regard to sensitivity analyses and interval estimates. I agree with the author's point that "a network modelling approach would be more appropriate for studying superspreading events and individual-level variation," thought it was appropriately placed in the Discussion

Reviewer #3 (Remarks to the Author):

The rebuttal the authors provided is comprehensive and addresses many of my questions: it is a certainly fair point that no model can answer every question, and there will be unavoidable uncertainty in some parameters. Furthermore, as the authors note, new data has come out that have supported some of the model's assumptions (such as mask efficacy and asymptomatic transmissibility). That said, it seems relatively few changes were made to the manuscript itself, and in my view there are still some remaining cases where further detail in the manuscript would be of value to readers. Specifically:

1. "indeed a number of insights from this study have proven correct. For example, our analysis highlighted the importance of communal accommodation and the higher infection rates in first year students in advance of the university outbreaks." -- I understand "have proven correct" to mean that some (presumably preliminary) data are available to validate the results of this study. If so, this would go a long way towards addressing many of my remaining concerns. As is, however, the most relevant passage in the manuscript I can find (noting that I could be missing a better example) is "Halls of residence have been identified as a risk factor for the transmission of close contact infections including meningitis[32], mumps[33], norovirus[34], respiratory illnesses[35] and gastroenteritis[36]." I want to echo the first reviewer's point that it is not reasonable to expect the model to be repeatedly revised in response to the latest data, especially post-submission, as this process could go on indefinitely. However, I think if preliminary data are available at the time of revision, including at least a qualitative discussion of them would be valuable.

2. "This value is reflective of the UK data, where the average time between symptom onset and test date is just over 2 days." -- If this value is based on data that of course would be ideal and would address my concern with this assumption. However the manuscript still says simply "We assumed that the university would be operating within Public Health England (PHE) guidelines, i.e. that symptomatic cases should be tested and self-isolate within 48 hours." If it is based on data, it should state the source of that data in the manuscript. If it is based on a guideline, a citation should probably still be given (the only thing I can find, again noting my limited familiarity with the England context, is

<https://www.gov.uk/get-coronavirus-test>, which says only "You need to get the test done in the first 8 days of having symptoms.").

3. "The interventions considered were guided by the university based on what was feasible." -- To me, this is also important information, as it means it is not "just" a modeling study but rather an exploration of a concrete policy option under consideration. The consultation with the university does not seem to be mentioned in either the methods or the acknowledgements. While a more detailed explanation may be preferable, even just including this sentence from the rebuttal in the manuscript would probably suffice.

4. "Halls accommodation is mostly limited to first year students and students in higher years tend to live in private rented houses that are much more dispersed through the city." -- That makes sense, but I suppose now I'm confused why there isn't a bigger difference between first year and subsequent years. I think being able to see the actual distributions of contacts, e.g. in the supplementary material, would be quite useful (e.g., what fraction are at the 24-person maximum size?).

5. "This is due to the fact that accommodation drives epidemics, but practical constraints mean that reducing living circle size sufficiently would not be implementable." -- I realize I misunderstood this before; apologies (I missed the word "between"). I am still a little confused though; in the model, under all scenarios, there is zero contact between different living circles, correct? I am not sure I understand how the modeling construct of a "living circle" (which I interpret to be a proxy for the fact that 200+ people will not all be having high levels of contact with each other, even if they share kitchens, bathrooms etc.) translates to the real-world concept of "maintaining social distancing" between them. If, as with several of the previous points, this example is based on a real intervention under consideration, I believe this should be explained in the manuscript.

6. "We have received positive comments about these figures from policy makers and stakeholders. The mixing matrix is sorted by year group, highlighting the increased contacts among first years." -- I was referring to sorting the rows and columns of panels (a)-(c) from largest to smallest so the vertical bands (i.e., economics and humanities) aren't the most prominent feature of the graphs. (If the matrices are sorted in a particular order, perhaps just an explanation of what this sorting is would help; to me it looks unsorted). I am also now confused why the matrices aren't symmetric. For example, in panel A what does it mean that point [Law, Economics] (top middle) is high but [Economics, Law] (middle right) is low? Final trivial point, if the scale bar on panel (d) could match the others (i.e. have log-spaced ticks rather than log-unit labels) it would be helpful -- it took me a while to realize that the upper limit of panel (d) matches the upper limit of panel (a).

Several additional points the authors may wish to consider:

1. The author's last comment, about "positive comments ... from policy makers and stakeholders", suggests that this study has been used for real-world decision making. I had not realized this; if possible, it would be a very valuable addition to the manuscript to describe this in more detail (e.g., if the stakeholders were involved in the formulation of the model or questions, if any recommendations were adopted etc).

2. The opening sentence about "are due to start ... September 2020" could be updated.

3. Previously the authors had been consistent with using 2 significant figures for results, which seemed appropriate to me. Here there are a few instances where there seem to be more (e.g. "4178 (98%P.I.: 1792 - 9804)").

4. Typo, line 266: "from 20 students to by 25% and 75%"

REVIEWER COMMENTS

Reviewer #1 (Remarks to the Author):

I am content with the revisions made by the authors, and believe that (subject to the other reviewers being satisfied) this work is suitable for publication.

I am pleased that the authors are willing to generate a persistent zenodo link for the code commit that will accompany the final version of the manuscript, and cheerfully remind them to do so when the time comes. As a regular user of code from published manuscripts I can attest that knowing exactly the version used to produce the results in the manuscript is very valuable. Thank you!

Thanks. We have released a version [doi:10.5281/zenodo.4759117](https://doi.org/10.5281/zenodo.4759117).

Reviewer #2 (Remarks to the Author):

I believe that the authors have done an excellent job in responding to reviewer comments—particularly with regard to sensitivity analyses and interval estimates. I agree with the author's point that "a network modelling approach would be more appropriate for studying superspreading events and individual-level variation," thought it was appropriately placed in the Discussion

Thanks.

Reviewer #3 (Remarks to the Author):

The rebuttal the authors provided is comprehensive and addresses many of my questions: it is a certainly fair point that no model can answer every question, and there will be unavoidable uncertainty in some parameters. Furthermore, as the authors note, new data has come out that have supported some of the model's assumptions (such as mask efficacy and asymptomatic transmissibility). That said, it seems relatively few changes were made to the manuscript itself, and in my view there are still some remaining cases where further detail in the manuscript would be of value to readers. Specifically:

1. "indeed a number of insights from this study have proven correct. For example, our analysis highlighted the importance of communal accommodation and the higher infection rates in first year students in advance of the university outbreaks." -- I understand "have proven correct" to mean that some (presumably preliminary) data are available to validate the results of this study. If so, this would go a long way towards addressing many of my remaining concerns. As is, however, the most relevant passage in the manuscript I can find (noting that I could be missing a better example) is "Halls of residence have been identified as a risk factor for the transmission of close contact infections including meningitis[32], mumps[33], norovirus[34], respiratory illnesses[35] and gastroenteritis[36]." I want to echo the first reviewer's point that it is not reasonable to expect the model to be repeatedly revised in response to the latest data, especially post-submission, as this process could go on indefinitely. However, I think if preliminary data are available at the time of revision, including at least a qualitative discussion of them would be valuable.

We have added a reference to a serosurvey of university students that found higher seropositivity in first year undergraduates and students living in halls of residence. We have added this text to the discussion, paragraph 2:

“Since we first conducted this work in May 2020, a cross-sectional serosurvey of students in five English universities found that first year undergraduates were three times more likely to be seropositive than other year groups and that seropositivity was 49% in students living in halls of residence [26].”

2. "This value is reflective of the UK data, where the average time between symptom onset and test date is just over 2 days." -- If this value is based on data that of course would be ideal and would address my concern with this assumption. However the manuscript still says simply "We assumed that the university would be operating within Public Health England (PHE) guidelines, i.e. that symptomatic cases should be tested and self-isolate within 48 hours." If it is based on data, it should state the source of that data in the manuscript. If it is based on a guideline, a citation should probably still be given (the only thing I can find, again noting my limited familiarity with the England context, is <https://www.gov.uk/get-coronavirus-test>, which says only "You need to get the test done in the first 8 days of having symptoms.").

We calculated the delay from the UK line list data and reference a study that uses the data. The statement in the manuscript now reads:

“We assumed that symptomatic cases should be tested and self-isolate within 48 hours, which is consistent with the median time between symptom onset and test of two days in the UK[25].”

3. "The interventions considered were guided by the university based on what was feasible." -- To me, this is also important information, as it means it is not "just" a modeling study but rather an exploration of a concrete policy option under consideration. The consultation with the university does not seem to be mentioned in either the methods or the acknowledgements. While a more detailed explanation may be preferable, even just including this sentence from the rebuttal in the manuscript would probably suffice.

We have added this statement to the manuscript:

“We focussed on interventions that could be implemented on top of wider control measures and were guided on feasibility by the UoB Scientific Advisory Group”

4. "Halls accommodation is mostly limited to first year students and students in higher years tend to live in private rented houses that are much more dispersed through the city." -- That makes sense, but I suppose now I'm confused why there isn't a bigger difference between first year and subsequent years. I think being able to see the actual distributions of contacts, e.g. in the supplementary material, would be quite useful (e.g., what fraction are at the 24-person maximum size?).

We have included a new section in the supplementary material with a distribution of household sizes and statistics associated with living circles.

New section in supplementary material:

Living circles

We assumed that students could transmit infection to other individuals in their living circles. Living circles were defined as other students living at the same postcode, or where the number of students at the same postcode was greater than the maximum living circle size (24 in the baseline case), we randomly allocated students at that postcode to a living circle. Figure S2 shows the number of students living at a single postcode and whether a halls of residence was located at that same postcode. The average number of students living at a single postcode for private accommodation was

3.25, and for halls of residence was 137. 3.5% of postcodes were associated with more than 24 students; these could have been halls of residence or private houses near each other.

Figure S2: The number of students living at a single postcode from UoB data for 2019/2020, and whether a halls of residence was located at that same postcode.

5. "This is due to the fact that accommodation drives epidemics, but practical constraints mean that reducing living circle size sufficiently would not be implementable." -- I realize I misunderstood this before; apologies (I missed the word "between"). I am still a little confused though; in the model, under all scenarios, there is zero contact between different living circles, correct? I am not sure I understand how the modeling construct of a "living circle" (which I interpret to be a proxy for the fact that 200+ people will not all be having high levels of contact with each other, even if they share kitchens, bathrooms etc.) translates to the real-world concept of "maintaining social distancing" between them. If, as with several of the previous points, this example is based on a real intervention under consideration, I believe this should be explained in the manuscript.

There is contact between living circles – via the university contacts and the random other contacts.

The query is regarding this sentence in the discussion: "Maintaining social distancing between living circles within residences is paramount for maintaining COVID-19 control". To avoid confusion, we have re-worded this to be:

"Limiting transmission within residences is paramount for COVID-19 control in university settings."

6. "We have received positive comments about these figures from policy makers and stakeholders. The mixing matrix is sorted by year group, highlighting the increased contacts among first years." -- I

was referring to sorting the rows and columns of panels (a)-(c) from largest to smallest so the vertical bands (i.e., economics and humanities) aren't the most prominent feature of the graphs. (If the matrices are sorted in a particular order, perhaps just an explanation of what this sorting is would help; to me it looks unsorted). I am also now confused why the matrices aren't symmetric. For example, in panel A what does it mean that point [Law, Economics] (top middle) is high but [Economics, Law] (middle right) is low? Final trivial point, if the scale bar on panel (d) could match the others (i.e. have log-spaced ticks rather than log-unit labels) it would be helpful -- it took me a while to realize that the upper limit of panel (d) matches the upper limit of panel (a).

The order of the rows and columns matrices was by faculty. We have reordered based on total contacts and changed the scale bar on panel (d) as suggested.

The matrices are not symmetric because they are divided by the number of students in each group – this is illustrated with an example in the supplementary material (see figure S1).

Several additional points the authors may wish to consider:

1. The author's last comment, about "positive comments ... from policy makers and stakeholders", suggests that this study has been used for real-world decision making. I had not realized this; if possible, it would be a very valuable addition to the manuscript to describe this in more detail (e.g., if the stakeholders were involved in the formulation of the model or questions, if any recommendations were adopted etc).

Thanks. We have added this statement:

"This work was provided to the Scientific Pandemic Influenza Group on Modelling (SPI-M) which provides modelling evidence to the UK government [37] and used directly to inform planning for re-opening at the University of Bristol."

2. The opening sentence about "are due to start ... September 2020" could be updated.

We have rephrased this to

"Despite the on-going COVID-19 epidemic, social distancing measures in many countries were beginning to be relaxed and universities across the world re-opened for the new academic term in September 2020", as well as updating other time-specific phrases.

3. Previously the authors had been consistent with using 2 significant figures for results, which seemed appropriate to me. Here there are a few instances where there seem to be more (e.g. "4178 (98%P.I.: 1792 – 9804)").

Thanks, we have reverted to using 2 significant figures to report results.

4. Typo, line 266: "from 20 students to by 25% and 75%"

Corrected, thanks.

REVIEWERS' COMMENTS

Reviewer #3 (Remarks to the Author):

The revised manuscript addresses almost all of my concerns; there are just two small remaining issues:

- By my reading, Vusirikala et al. does not say "seropositivity was 49% in students living in halls of residence"; it looks to me like it was 28.9% (see Table 1). 49% was for a higher-risk subgroup.

- "We calculated the delay from the UK line list data and reference a study that uses the data. The statement in the manuscript now reads: "We assumed that symptomatic cases should be tested and self-isolate within 48 hours, which is consistent with the median time between symptom onset and test of two days in the UK[25]."" -- This sounds fine, but I don't actually see this change in the manuscript. It looks like the text in the manuscript still reads "assuming that symptomatic cases are tested and self isolate within 48 hours."

REVIEWERS' COMMENTS

Reviewer #3 (Remarks to the Author):

The revised manuscript addresses almost all of my concerns; there are just two small remaining issues:

- By my reading, Vusirikala et al. does not say "seropositivity was 49% in students living in halls of residence"; it looks to me like it was 28.9% (see Table 1). 49% was for a higher-risk subgroup.

We have clarified this statement in the discussion to:

"seropositivity was 49% in students living in halls of residence in universities with high SARS-CoV-2 infection rates"

- "We calculated the delay from the UK line list data and reference a study that uses the data. The statement in the manuscript now reads: "We assumed that symptomatic cases should be tested and self-isolate within 48 hours, which is consistent with the median time between symptom onset and test of two days in the UK[25]."" -- This sounds fine, but I don't actually see this change in the manuscript. It looks like the text in the manuscript still reads "assuming that symptomatic cases are tested and self isolate within 48 hours."

Thank you – this sentence must have slipped through from a previous version. We have now corrected this.